

# A versatile water vapor generation module for vapor isotope calibration and liquid isotope measurements

Hans Christian Steen-Larsen[1], Daniele Zannoni[1,2]

[1]Geophysical Institute, University of Bergen and Bjerknes Centre for Climate Research, Bergen, Norway
[2]Department of Environmental Sciences, Informatics and Statistics, University Ca' Foscari of Venice, Venice, Italy

*Correspondence to*: Hans Christian Steen-Larsen ([Hans.Christian.Steen-Larsen@uib.no](mailto:Hans.Christian.Steen-Larsen@uib.no))

**Abstract.**

A versatile vapor generation module has been developed for the purpose of both field water vapor isotope calibrations and laboratory liquid water isotope measurements. The vapor generation module is fully scalable allowing in principle an unlimited number of standards or samples to be connected, opening up the possibility for calibrating with multiple standards during field deployment. Compared to a standard autosampler system, the vapor generation module has a more than 2 times lower memory effect. The vapor generation module can in principle generate a constant stream of vapor with constant

isotopic composition indefinitely. We document an Allan Deviation for $^{17}O$-excess ($\Delta^{17}O$) of less than 2 per meg for an approximate 3 hour averaging time. For similar averaging time the Allan Deviation for $\delta^{17}O$, $\delta^{18}O$, $\delta D$, d-excess is 0.004, 0.006, 0.01, 0.03 ‰. Measuring unknown samples show that it is possible to obtain an average standard deviation of 3 per meg leading to an average standard error (95 % confidence limit) using 4-5 replicates of 5 per meg.

Using the vapor generation module we document that an enhancement in the Allan Deviation above the white noise level for

integration times between 10 minutes and 1 hour is caused by cyclic variations in the cavity temperature. We further argue that increases in Allan Deviation for longer averaging times could be a result of memory effects and not only driven by instrumental drifts as it is often interpreted as.

The vapor generation module as a calibration system have been document to generate a constant water vapor stream for a period of more than 90 hours showing the feasibility of being used as an autonomous field vapor isotope calibration unit for

more than 3 months.

## 1 Introduction

Water samples from the atmospheric hydrological cycle in the state of liquid, solid, and vapor offer an extraordinary tool to understand how meteorological and hydrological processes drive the climate system. The isotopic composition of meteoric water represents integrated information of the water cycle from evaporation at the ocean and land surface along the air mass

trajectory until the water molecules in the end falls back to the surface either as liquid or solid precipitation (e.g. Galewsky et al., 2016).





The relative abundance of HD$^{16}$O and H$_2$$^{18}$O compared to H$_2$$^{16}$O of meteoric water samples have routinely been measured for the last more than 60 years (e.g. Craig and Gordon, 1965; Dansgaard, 1964) and in recent decades also along with the relative abundance of H$_2$$^{17}$O (e.g. Steig et al., 2014; Luz and Barkan, 2005). The relative abundance of HD$^{16}$O, H$_2$$^{17}$O, H$_2$$^{18}$O compared to H$_2$$^{16}$O, referred to using the δ-notation (Craig, 1961), are classically understood through the first order principle of temperature driven distillation during transport in the atmosphere (Dansgaard, 1964). To account for kinetic processes during phase transition for example during ocean evaporation (Merlivat and Jouzel, 1979), formation of snow crystals (Jouzel and Merlivat, 1984), sublimation from snow (Wahl et al., 2021), and plant transpiration (Landais et al., 2006), second order parameters such as the d-excess (Dansgaard, 1964) and the $^{17}$O-excess ($\Delta^{17}$O) (Landais et al., 2008) are defined as:

$$d - excess = \delta D - 8 \times \delta^{18}O$$

$$\Delta^{17}O = Ln(\delta^{17}O + 1) - 0.528\, Ln(\delta^{18}O + 1)$$

The number of atmospheric water vapor isotope measurements have over the last decade increased rapidly thanks to the availability of commercial water vapor isotope laser spectroscopy analysers. Similarly, the number of laboratories that carry out routine measurements of liquid water samples for their isotopic composition has increased thanks to the lowering in costs of acquiring a laser spectroscopy analyser and the ease of use compared to IRMS analysers. This development has allowed the scientific community to enhance the understanding of the hydrological cycle by not only using collected precipitation samples, but also measuring the exchange and transport of water vapor in the climate system. However, the availability of high accuracy water vapor isotope measurements has hinged on the ability to make robust and reliable field measurements of water vapor with known isotopic composition based on laboratory standards referenced against the international VSMOW-SLAP scale. Similarly, the availability of second order parameters such as the d-excess or the $\Delta^{17}$O are dependent on the ability to make high accuracy measurements of $\delta^{18}$O, $\delta^{17}$O, and $\delta$D with a relatively high output of individual number of samples measured.

Commercial systems including the Water Vapor Isotope Standard Source (WVISS) from Los Gatos Research and the Standard Delivery Module (SDM) from Picarro inc. have been developed together with commercial water vapor isotope analysers. However, challenges in calibration of measurements using these systems for long deployments (Bonne et al., 2014; Steen-Larsen et al., 2015) or deployments in low humidity regions (Guilpart et al., 2017) have been documented. Custom-made changes to the SDM system have been carried out to facilitate long-term deployments (Bastrikov et al., 2014). While the SDM system allows two standards to be used, the WVISS system only allows a single standard to be measured without manually changing the standard. For the WVISS this implies that a VSMOW-SLAP scale calibration cannot be performed automatically. For the SDM this implies that the accuracy of the calibration cannot be assessed using a third known standard as is standard protocol for liquid water isotope measurements (e.g. Van Geldern and Barth, 2012).


To overcome some of these challenges custom-made calibration systems have been developed, specifically targeting the conditions of the field deployment. The simplest and to date most robust calibration system in terms of long-term deployment is the so-called bubbler system. Such a system has been deployed continuously at the water vapor isotope

monitoring station in Bermuda for more than 10 years (Zannoni et al., 2022; Steen-Larsen et al., 2014). The versatility of the system lies in the wide humidity range (1000 ppm to 30 000 ppm) and that there is minimal need for manual intervention to operate (e.g. Bailey et al., 2015; Ellehoj et al., 2013). However, for the system to operate with low uncertainty it is a requirement that temperature stability to within half a degree is obtained. In addition, large quantities of water standards are needed and it is not practical to transport. These constraints means that the bubbler system is not always feasible to be used

during most field campaigns.

Besides bubbler-systems, custom-made calibration systems have been developed on the principle of either complete evaporation of small water droplets or a steady-state evaporation of a sub-millimeter size droplet. To the former group of calibration systems counts the use of dew point generator (Lee et al., 2005), dripper-systems (Tremoy et al., 2011; Lee et al.,

2005), nebulizer-systems (Jones et al., 2017), and piezoelectric microdrop generators (Iannone et al., 2009). While these systems have been proven to work under field conditions, they require significant time to equilibrate and may potentially be sensitive to room temperature fluctuations. For the operation of the nebulizer-system, the system needs in the order of 5 L/min dry air to achieve low humidity levels, which provide additional need for bringing a high performing dry air generator into the field, which add logistical challenges of organizing field campaigns.


To the group of calibration systems operating on the principles of steady-state evaporation belongs a field deployable system developed by Gkinis et al. (2011) for continuous ice core water isotope analysis directly in the field and a dedicated low-humidity calibration system developed by  Leroy-Dos Santos et al. (2021). The system developed by Gkinis et al. (2011) used a combination of a peristaltic pump and tuned back-pressure to push water through a silica capillary into a micro-Tee

heated to 170 deg C.  A similar system has also been deployed and used for calibration of water vapor isotope measurements (Benetti et al., 2017; Steen-Larsen et al., 2014). This system has also shown to be reliable for 17O-excess measurements in continuous-flow analysis of ice core samples (Davidge et al., 2022).

For isotope measurements carried out on atmospheric water vapor and liquid samples, it is crucial in order to achieve data of

highest quality to address sources of measurement uncertainty arising through both measurement protocol and instrumental drift. Several published protocols exist for liquid samples (e.g. Hutchings and Konecky, 2023; Penna et al., 2012; Van Geldern and Barth, 2012) and atmospheric water vapor (e.g. Bailey et al., 2015; Steen-Larsen et al., 2013) along with a multitude of laboratory specific protocols. However, improvement of liquid measurements protocols for better precision and accuracy has been restricted to adjusting the number of injections, the amount of injected liquid, and to a small degree





changes in integration time. For measurement protocols of water vapor isotopes, improvements have been restricted by the lack of a possibility of generating a continuous vapor stream with constant and known isotopic composition at different humidity levels using more than 2 standards.

For the purpose of not only developing a field deployable water vapor isotope calibration system targeting the short-comings
described above, but also developing a system for liquid sample measurements exceeding available systems in throughput and accurateness we have developed further the patent application (Steen-Larsen, 2016). Our target for the water vapor isotope calibration needs were:
- Field deployable i.e. robust, transportable, and easy to operate
- Multiple standard measurement capability
- High humidity span from 300 to 30 000 ppmv
- High flow rate span of vapor with a constant known isotopic composition
- Possibility to calibrate with sufficient high accurateness to obtain $\Delta^{17}O$ data

Our target for the liquid measurements were:
- Scalable system in terms of number of unknown samples
- Improvement in sample measurement time compared to discrete injections for similar precision
- User defined choice of integration time and hence measurement accurateness and precision.
- Comparable sample material use as discrete measurements/Small quantity of sample material required.

The developed system should have the ability to be used both for liquid measurements and for calibration of water vapor isotope measurements. In addition, the system should have sufficient stability to function as a benchmark tester for isotope analyser performance by supplying a constant stream of water vapor with a known isotopic composition over multiple days.

To further demonstrate the fulfilment of all the proposed targets, we report two case studies as representative of water
isotope calibration and liquid measurements applications:
- A humidity-isotope characterization curve performed during a field campaign in the arctic,
- The liquid analysis of $\Delta^{17}O$ of artificial samples generated from mixing of two reference waters with known $\Delta^{17}O$ values.





## 2 Materials and Methods

### 2.1 Vapor Generation Module description

The vapor generation module is an improved and revised version of an original prototype developed in 2014 for which a patent application was filed in 2015 (Steen-Larsen, 2016). The previous prototype was reengineered at the beginning of 2021 and its schematic with a generic number of ovens ($n$) is reported in Fig. 1. A four-ovens version was used in this study.

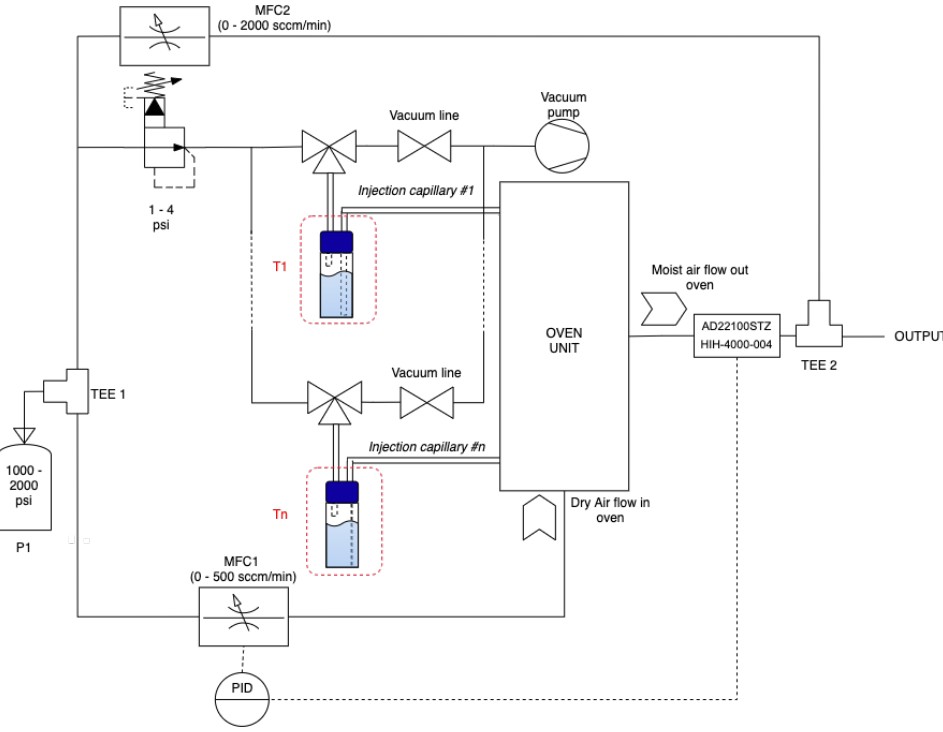

**Figure 1: Sketch of the vapor generation module equipped with multiple ovens. The gas analyser must be connected at the OUTPUT of the vapor generation module in open-split mode.**

The operating principle of the vapor generation module is based upon the *flash evaporator* method, which is adopted in the water stable isotopes community for continuous flow analysis of ice-cores and for atmospheric water vapor analysis (e.g. Gkinis et al., 2011; Leroy-Dos Santos et al., 2021). In the flash evaporator, a droplet of water continuously evaporates on the tip of a needle/capillary inside an evaporation chamber at high temperature. The temperature of the evaporation chamber is precisely controlled well-above the boiling point (in this work at 170°C) and the volume is continuously flushed with dry air to produce a water vapor stream with the same isotopic composition of the water source in the needle. As recently demonstrated, no fractionation occurs during operation in steady state and back diffusion from the tip through the capillary line is not sufficient to affect the water vapor isotopic composition inside the water reservoir (Kerstel, 2021; Leroy-Dos Santos et al., 2021). To push the water through the capillary, previous studies used syringe pumps or combinations of tubings of different size to generate sufficient back pressure (Gkinis et al., 2011; Landsberg et al., 2014; Leroy-Dos Santos et al.,



2021). In this study we used compressed dry air, with headspace pressure precisely controlled by an electronic pressure regulator (ALICAT PCD-5PSIG, resolution 0.01 PSI), as proposed in Steen-Larsen (2016).


Main improvements from the original design are the presence of:

1. A vacuum pump that can be connected to the vial headspace via three-way valve and a vacuum valve to halt vapor generation in the oven.
2. A PID control loop to control and stabilize humidity level.
3. An additional mixing tee connected (TEE2) to a secondary MFC (MFC2) to ensure larger dynamic humidity range.
4. The use of a dual valve pressure controller flow meter to enable seamless control of humidity level.
5. Use of stainless steel capillaries instead of fused silica capillary tube.
6. The option of using a single oven connected to a selector valve connected to individual vials.

These main improvements compared to the original patent application (Steen-Larsen, 2016) were implemented with the following operations in mind: (1) the purpose of the vacuum pump is to create a small vacuum in the vials' headspace, which reverses the flow of the water in the capillaries out of each oven and back into the vials. The benefit of this is that water is immediately removed from the capillary and the oven dries out immediately, minimizing cross contamination of the samples and build-up of water inside the oven. (2) A custom RH/T probe is installed in the tube where the vapor is flowing to the 165 analyser and internally looped to the primary MFC to control the humidity level. This is necessary because of potential for slowly decreasing trend of humidity during long injection time (several hours). The decrease of injection performance during long runs is attributed to build-up of salt deposits inside the capillary during evaporation (further discussed in Section 4.2). (3) The secondary MFC allows for a second dry air stream to be mixed into the flow of vapor from the oven at a higher flow rate than the primary MFC and without influencing the production of vapor in the oven. This way, the vapor generation 170 module can generate humidity levels at a much lower level than possible using only mixing in the oven setup and can also generate vapor with a higher flow rate. (4) The dual valve pressure controller flow meter installed to regulate the pressure in the vials' headspace is chosen to ensure that the pressure can both be increased and decreased. (5) As discussed in Section 4.2 one of the main issues with operation of a flash evaporator of this type is clogging of the capillaries. We observed that the use of stainless steel capillaries reduced the frequency of clogging compared to when using fused silica capillaries. (6) 175 Instead of having individual oven and valve circuits for each sample vial we installed a selector valve similarly to Jones et al. (2017) for the purpose of cost optimization. Unfortunately, the use of a selector valve comes at the cost of increased memory effect between samples and hence we opt for a combination of individual oven and valve circuits and the use of a 10-port selector valve. For generating a stream of water vapor from standards we use the individual oven and valve circuits while liquid samples are measured using the selector valve and single oven circuit.




## 2.2 Technical realization of vapor generation module

For the technical realization of the vapor generation module, a 2 ml sample vial commonly used in autosamplers for liquid water isotope analysis was used as the water reservoir, as shown in Fig. 2 panel a. A PTFE capillary (1.59 mm OD, 0.150 mm ID) was passed through a tee assembly and 3.18 mm OD stainless steel tubing (ID > 1.59 mm) toward the bottom of the vial which was kept in place with a Swagelok UltraTorr fitting. The PTFE capillary is sealed only at the very top of the tee assembly with a PTFE ferrule and a PTFE nut. The pressure inside the tee assembly is equal to the pressure in the vial headspace and can be precisely controlled with a combination of a manual pressure regulator (model 8286, Parker) and of an electronic dual valve pressure controller (PCD-5PISG-D-SV, ALICAT), which includes an advanced PID control of the pressure. Any change of the pressure in the headspace (positive or negative) will produce a flow of water through the PTFE capillary. In this study, the headspace pressure was regulated between 0.5 to 3.5 PSI.

The PTFE capillary is connected with a 1.59 mm to 1.59 mm PEEK union to a 100 mm long stainless steel capillary with 0.127 mm ID (T10C5, VICI) and fitted into one end of a stainless steel tee (ZT1M, VICI) which is kept in place over a 200W heating element (860-6912, RS-PRO) with a custom aluminum assembly, as shown in Fig 2 panel b. The stainless steel tee and the heating block are placed into an aluminum diecast enclosure (Hammond Mfg.) and thermally insulated with machinable glass ceramic rods (Cornic) and wrapped in fiberglass liner. As mentioned before, in this study we used a 4 elements version of the oven, but the module is highly scalable in terms of number of oven units.

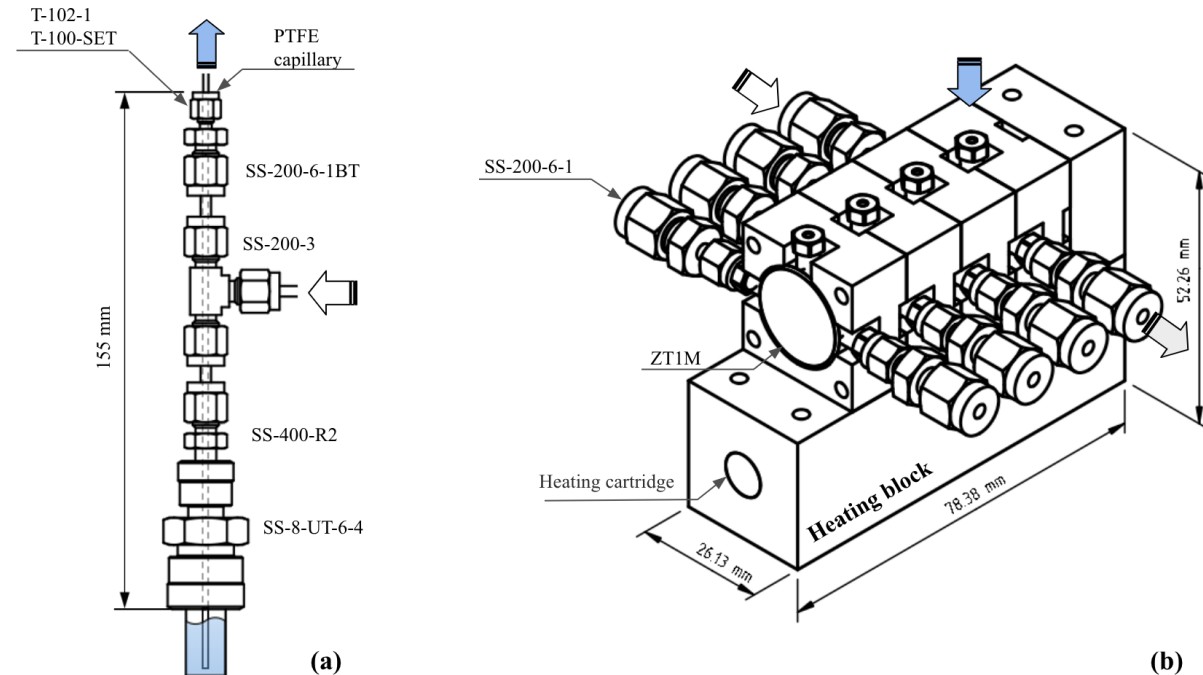

(a)                                                                                                          (b)





**Figure 2: Technical drawings of the components used to produce the water vapor stream. (a) Detail of the custom vial holder (Swagelok part numbers). (b) Detail of the compact 4 ovens assembly (Swagelok and VICI part numbers). For both panels, white arrows indicate push gas/dry air, blue arrows indicate water flow, gray arrow indicates moist air flow.**

The water vapor produced in the oven is routed toward the output of the system where a custom RH/T probe is used to monitor the temperature and the humidity of the outflow gas (AD22100STZ, Analog Device and HIH-4000-004, Honeywell inside a SS-400-3-4TTF tee, Swagelok). The RH/T probe is used to control the flow rate of the first MFC (GFC17A, 0-500 ml/min, AALBORG) with a PID, in order to keep a steady humidity level at the output of the vapor generation module during long injections. A data acquisition module (USB-6001, National Instrument) was used to control valves and MFCs. The control software was written in LabView 16.

### 2.2.1 Continuous injections of different samples

In a second modification of the vapor generation module, we used a single oven connected with the stainless steel capillary to a multiport selector (C25-3180EUHA, VICI) to inject samples from the different vial holders continuously, without needing to switch the oven. This second modification was introduced to analyse samples of unknown isotopic composition in a similar fashion to CFA. The possibility of changing between water sources without affecting the water vapor production enables both measurements of individual unknown samples and the possibility of running long stability tests (>24h) since switching between samples containing the same standard occur seamlessly.

### 2.3 Standards and samples used for testing the vapor generation module

A wide range of isotopic values were used to test the calibration module, spanning approximately the VSMOW-SLAP2 range (Coplen, 1988; Gonfiantini, 1978). Laboratory standards with known isotopic composition for this study were provided by Laboratoire des Sciences du Climat et de l'Environnement, Centre for Ice and Climate at the Niels Bohr Institute, and the Stable Isotope Laboratory at the Institute of Arctic and Alpine Research, University of Colorado. The 17O-excess values of the standards provided by the Stable Isotope Laboratory were provided by Δ*IsoLab, University of Washington. An overview of all the standards used are listed in supplementary material Table S1, while Table 1 shows the most frequent used standards for analysing stability and memory effects. Moreover, four more samples (Supplementary material Table S2) were prepared at the University of Bergen by weighting precise amounts of SW and WW standards with an uncertainty of 0.01g to test the reproducibility of the $\Delta^{17}O$ measurements. For these four samples, the uncertainty associated to $\Delta^{17}O$ value is assumed to be half the maximum span obtained by mixing SW +/- 0.01g with WW -/+ 0.01g, which is approximately 0.013 ‰ (13 per meg).



**Table 1: Standards used for stability and memory effect analysis in this study. Precision (±) is the standard error.**

| Name | $\delta D$ (‰) | $\delta^{17}O$ (‰) | $\delta^{18}O$ (‰) | d-excess (‰) | $\Delta^{17}O$ (per meg) |
|------|------|------|------|------|------|
| BER | $-2.10 \pm 0.17$ | $-0.05 \pm 0.02$ | $-0.25 \pm 0.02$ | 4 | 82 |
| SP | -435.31 | -29.6497 | -55.39 | 8 | -11 |

**2.4 Isotopic water vapor analyzer**

Most of the laboratory characterization tests for the vapor generation module were performed using a L2140-i Cavity Ring-
Down Spectroscopy (CRDS) water isotope analyzer by Picarro (s.n. HKDS2156). In one occasion, to identify the source of
measurement noise, the analyzer was run simultaneously with another Picarro L2140-i (s.n. HKDS2092). The HKDS2092
model was also used for the field activity mentioned in this work (see Section 3.4). Chronologically, the HKDS2092 is the
oldest manufactured instrument that was used while HKDS2156 is the most recent one. Both the HKDS2092 and the
HKDS2156 have an acquisition rate of ~1 Hz, which is a typical value for this type of instrument. The reader is referred to
previous studies for detailed description of the technology and performances of such analyzers (e.g. Steig et al., 2014).

**3 Results**

**3.1 Long term injection and long term stability**

To demonstrate the long-term stability of the vapor generation system, a 92 hours test was performed during the fall 2022,
from Oct 10[th] 16:20 to Oct 14[th] 12:25 (LST). During the test, a 2 ml aliquot of SP standard (1 vial) and 26 ml aliquot of BER
(13 vials) were injected using the four available vial holders and routed to a single oven using a multiport selector. To
prevent contamination between SP and BER standards, the first holder was used only for SP standards while the remaining
three were cycled every 24 hours using fresh BER standards. Sequence, timing and relevant statistics of the injections are
reported in supplement material Table S3. The PID control loop was set to 17300 ppmv and the final averaged humidity
recorded was $17229 \pm 340$ ppmv with no observable trend, as shown in Fig. 3 panel a. For ~80% of injections the standard
deviation of $H_2O$ is between 113 - 353 ppmv, yielding an RSD in the range 0.7 - 2%. There is apparently no pattern in the
variability of $H_2O$ signal among the different holders. At the working humidity between 15000 and 20000 ppmv the
humidity-isotope response curve is almost flat for $\delta^{17}O$ and $\delta^{18}O$. However, a small dependency was observed between $H_2O$
and $\delta D$, as reported in supplementary material Fig. S1.

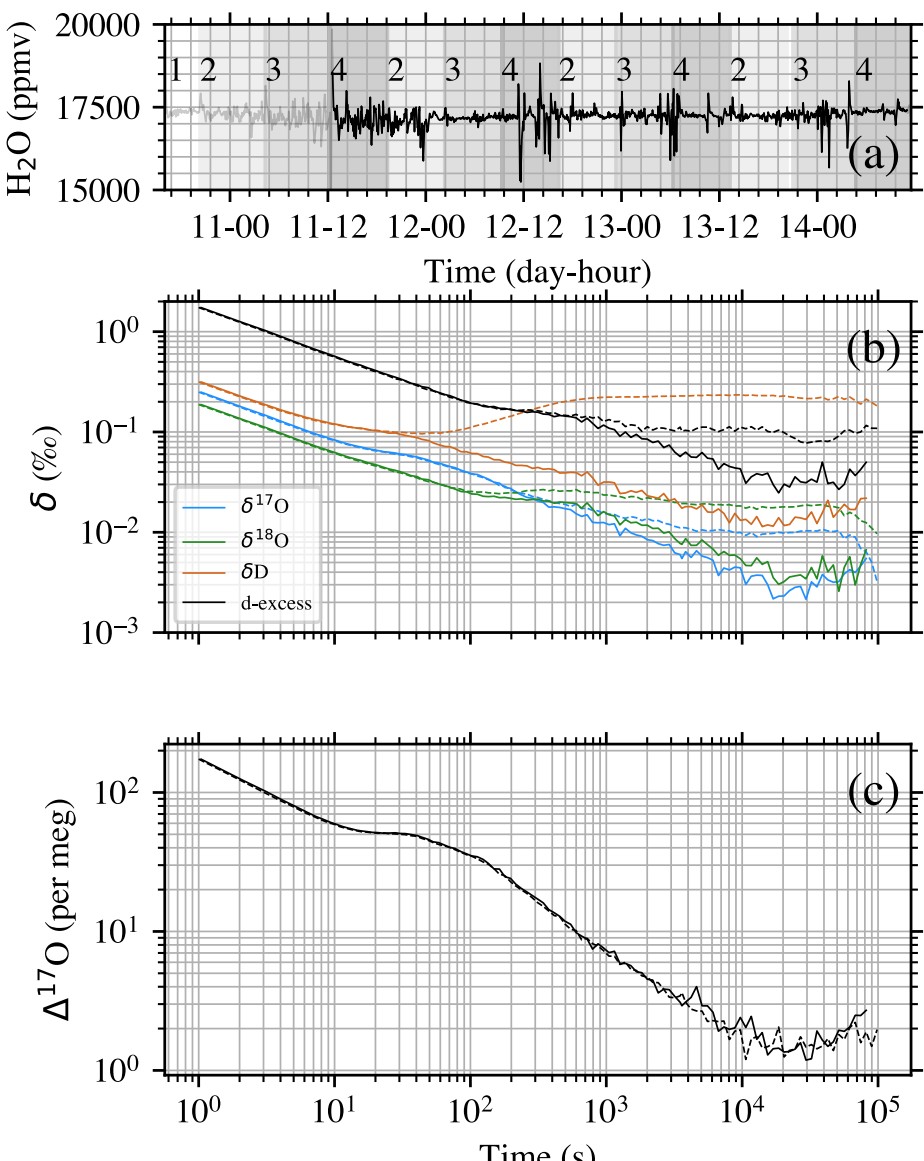

**Figure 3: Flash evaporator long stability test of BER standard. (a) Timeseries of $H_2O$ signal (17229 ± 340 ppmv). The signal used to calculate the Allan Deviation is reported as a solid black line. On the x-axis: day (October 2022) and hour (LST). Gray shading used to distinguish among injections from different vials. The ID of vial holders are reported on the top of the plot. (b) Allan Deviation of $\delta^{17}O$, $\delta^{18}O$, $\delta D$ and d-excess during the long-term stability test, including the full 92 hours dataset (dashed lines) and discarding the first 16 hours (solid lines). (c) Similar to b but for $\Delta^{17}O$.**

The Allan Deviation plots in Fig. 3 panel b and c (solid lines) show a constant improvement of the two-sample variance, consistent with a nearly flat noise spectrum (white noise). For ideal white noise, a slope of -0.5 $\sigma_{\text{Allan}}/\tau$ is expected. Observed slopes between averaging time 1 to $10^4$ seconds are -0.42, -0.33, -0.31 and -0.38 ‰/s for $\delta^{17}O$, $\delta^{18}O$, $\delta D$ and d-excess,





respectively, and -0.42 ppm/s for $\Delta^{17}O$. A slope smaller than the one predicted for pure white noise implies that the non-flat spectral characteristic of the measurement noise can be due to instability of the analyser as well as from instability of the

vapor generation module. We note that for some ranges of averaging times the measurement noise can be seen to be pure white noise. For example, white noise is observed for d-excess and $\Delta^{17}O$ in the ranges 1 to $10^2$ and 1 to $10^1$, and again in the ranges $10^3$ to $2 \cdot 10^4$ and $2 \cdot 10^2$ to $10^4$ seconds, respectively. In the range $\sim 10^2$ to $10^3$ and $\sim 10^1$ to $2 \cdot 10^2$ seconds for respectively d-excess and $\Delta^{17}O$ it can be noted that instabilities of the analyser exist, which has consequences for optimal choice of averaging time when weighing precision up against averaging time. It is worth noting that the Allan Deviation plot presented

in this study shows better isotopic analyser performances than previously published in other studies (e.g. Sturm and Knohl, 2010; Steig et al., 2014; Leroy-Dos Santos et al., 2021; Aemisegger et al., 2012), yielding 0.004, 0.006, 0.01, 0.03 ‰ $\sigma_{Allan}$ for $10^4$ seconds averaging time for $\delta^{17}O$, $\delta^{18}O$, $\delta D$, d-excess respectively and 2 per meg $\sigma_{Allan}$ for $10^4$ seconds averaging time for $\Delta^{17}O$. We argue that this apparent better performance to not only be dependent on the isotope analyser quality, even though instrumental performance improved with the introduction of the Picarro L2140-i series, but dependent on a

suboptimal management of memory effect during Allan Deviation tests. As mentioned above, the 92 hours test started with a large positive isotopic step change in water vapor source ($\sim 55$‰ for $\delta^{18}O$) and the following 16 hours were removed from the Allan Deviation analysis to minimize memory effect. If the extra 16 hours data is used to compute the Allan Deviation, the plateau is reached already at $4 \times 10^3$ s and the $\sigma_{Allan}$ is larger (dashed lines in Fig. 3 panel b). Most notably, an apparent drift effect shows up for $\delta D$ after an averaging time of $10^2$ s. Interestingly, the $\Delta^{17}O$ quantity seems to be unaffected by this

memory effects issue. To test the long-term stability and reproducibility of the Allan Deviation characterization, an independent characterization was carried out 8 months later. The test revealed a similar result as presented in Fig. 3.

### 3.2 Isotope step change and reduced memory effect

Assuming 24 hours of continuous injection of the same water standard are enough to completely remove the memory effect of the previous sample, we show in Fig. 4 the temporal evolution of the SP-BER normalized isotopic step change curve for

fractions of the normalized target value larger than 0.90. For comparison, we ran a similar step-change test using the liquid injection mode of the Picarro L2140-i instrument ($^{17}O$, high precision mode) coupled to the A0235 autosampler and A0211 vaporizer. For the liquid injections test the standards were analysed as follows: 2 vials of SP (25 injection each) followed by 9 vials of BER (25 injection each), for a total number of 275 injections and a running time of $\sim 41$ hours. Results clearly show that the vapor generation module is faster than the vaporizer using the autosampler system for reaching the target value

after the step change. For instance, using the vapor generation system the 0.995 level is reached after 24, 23 and 47 minutes for $\delta^{17}O$, $\delta^{18}O$ and $\delta D$, respectively. The same level is reached with the vaporizer using the autosampler system after 54, 54 and 152 minutes. However, the liquid injections with the vaporizer using autosampler are evenly spaced by cleaning cycles of the vaporizer, robotic arm movements, etc. All such operations require a considerable amount of time, which is mentioned hereafter as dead time. Fig. 4 shows that timings for calibration system and vaporizer using autosampler are comparable for

$\delta^{17}O$ and $\delta^{18}O$ when dead time is ruled out. However, when theoretically removing the dead time, the vaporizer using the





autosampler requires ~77 minutes to reach the 0.995 level of target value for $\delta D$, which is still considerably longer by about ~60% than the time required by the presented vapor generation system.

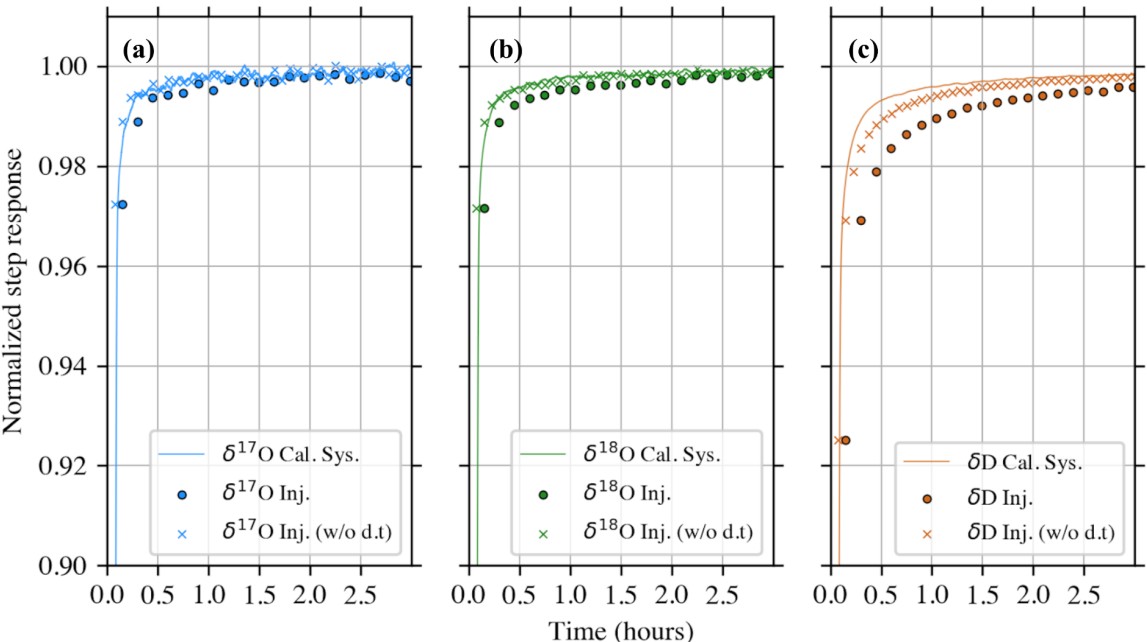

**Figure 4: Normalized step change between standard SP and BER for $\delta^{17}O$, $\delta^{18}O$, and $\delta D$ in (a), (b), and (c), respectively. Solid lines: continuous signal acquired from the calibration system (moving average 270s). Coloured circles: injections performed with the Picarro autosampler and vaporizer including dead time. Coloured crosses: injections performed with the Picarro autosampler and vaporizer removing dead time.**

### 3.3 Short term stability

The vapor generation module was tested by injecting the BER standard at 12 different humidity levels with injection times ranging from 3.5 to 8.7 hours (Table 2).

**Table 2: Humidity levels selected to test the short term performances of the vapor generation module. H2O $1\sigma_{1-s}$ is one standard deviation calculated at standard sampling rate (~1 Hz).**

| H$_2$O avg. (ppmv) | H$_2$O $1\sigma_{1-sec}$ (ppmv) | RSD (%) | $\Delta H_2O/\Delta t$ (ppmv/h) | Duration (h) |
|---|---|---|---|---|
| 584 | 44 | 7.5 | 2 | 6.1 |
| 995 | 30 | 3.0 | 11 | 4.9 |
| 2421 | 101 | 4.2 | 29 | 5.8 |





| | | | | |
|---|---|---|---|---|
| 4538 | 18 | 0.4 | 10 | 4.5 |
| 7026 | 82 | 1.2 | 24 | 7.3 |
| 8902 | 44 | 0.5 | 10 | 5.0 |
| 11001 | 47 | 0.4 | -1 | 6.9 |
| 11584 | 63 | 0.5 | -5 | 3.5 |
| 13856 | 54 | 0.4 | 24 | 5.6 |
| 16047 | 89 | 0.6 | -1 | 8.7 |
| 17902 | 89 | 0.5 | -18 | 3.8 |
| 19616 | 110 | 0.6 | -24 | 5.4 |

Such humidity levels cover almost completely the mixing ratio variability of the Earth's lower troposphere, with the exception of very dry regions like Antarctica (Casado et al., 2016), and warm-wet regions like tropical areas (e.g. Laskar et al., 2014) . The stability of water vapor signal was evaluated with three metrics for each level:

**Metric 1: The spread-variability of $H_2O$ signal.** Results in Table 2 show that the 1 second standard deviation ranges
between 18 and 110 ppmv and is independent from the humidity level. If a smaller variability is required, the vapor generation module can be locked to a humidity level and a second dry air flow can be used to dilute the water vapor concentration at TEE2 (Figure 1)**.** The application study in Section 3.4 reports an example of usage of the second stage mixing and the associated 1 second standard deviation between humidity levels 500 - 3500 ppmv.

**Metric 2: The short-term trend of the $H_2O$ signal ($\Delta H_2O/\Delta t$).** Results in Table 2 show that the trend, evaluated in 3.5 - 8.7 hours is limited in a narrow range from -24 ppmv/h to 29 ppmv/h. Positive trends were observed mostly below 9000 ppmv, while negative trends above 11000 ppmv. A linear fit of the observed trend vs humidity reveals that a trend of ~0 ppmv/h is achievable at ~12500 ppmv, which can be considered the most stable region of the calibration module in terms of drift of mixing ratio level.


**Metric 3: The overlapping Allan Deviation of $\delta^{17}O$, $\delta^{18}O$, $\delta D$ and $\Delta^{17}O$.** In general, a significant improvement of the two-sample variance can be observed above 5000 ppmv (Appendix 1 Figure A1). As expected, the largest Allan Deviation with averaging time of 600 seconds is the one measured at 584 ppmv level (0.05, 0.05 and 0.18 ‰ for $\delta^{17}O$, $\delta^{18}O$ and $\delta D$, respectively). Between 5000 and 20000 ppmv, the 600 s Allan Deviation is characterized by small variability (0.014 ± 0.002,
0.015 ± 0.004 and 0.03 ± 0.01 ‰ for $\delta^{17}O$, $\delta^{18}O$ and $\delta D$, respectively). For unknown reasons the worst performances in





terms of Allan Deviation are observed at 11584 ppmv, in contrast with the analysis above, which shows the smallest short-term trend in the ~12500 ppmv region. Similar to $\delta^{17}O$, $\delta^{18}O$ and $\delta D$, the overlapping Allan Deviation of $\Delta^{17}O$ follows in general the same pattern, as shown in Fig. 5. However, the decrease of performances observed for $\delta^{17}O$, $\delta^{18}O$ and $\delta D$ at 11584 ppmv is limited for $\Delta^{17}O$. The optimal working region of the Picarro L2140i coupled to the vapor generation module is between 14000 and 18000 ppmv.

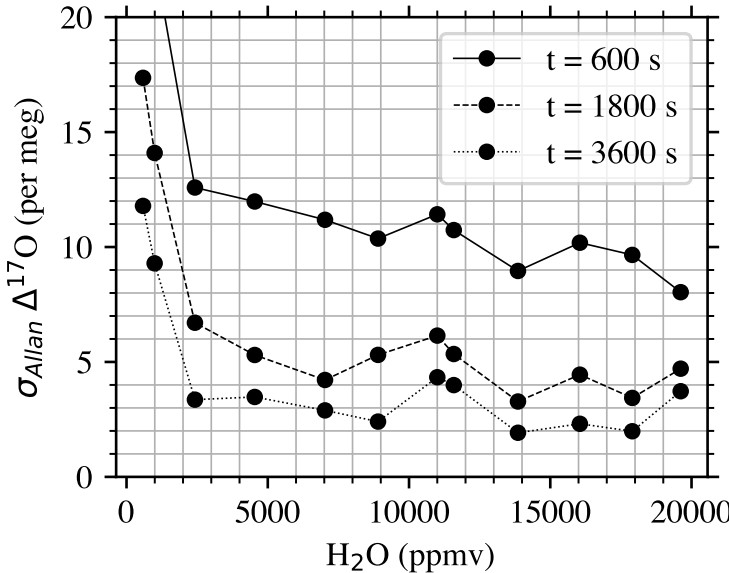

**Figure 5: Overlapping Allan Deviation of $\Delta^{17}O$ measured at different humidity levels and for different integration times.**

### 3.4 Application study 1: Humidity - isotope characterization of water vapor analyzer for the lower polar troposphere (WIFVOS)

Obtaining accurate vertically resolved profiles of water vapor isotopic composition are challenging because of the large isotopic and humidity span in the atmospheric column. This is even more extreme in the polar regions, where humidity can easily be < 1000 ppmv (e.g. Casado et al., 2016; Wahl et al., 2021). Here we show how the calibration system can be successfully used to fully characterize the humidity-isotope response of CRDS water vapor analysers between 500 - 3500 ppm, where commercially available calibration systems notoriously have issues generating stable humidity signal (Ritter et al., 2016; Guilpart et al., 2017). The data of this application study was acquired during the WIFVOS field campaign in Sodankylä, Finland, for the characterization of the HKDS2092 analyser used in the field to measure the isotopic composition of water vapor collected with flasks between ground level and ~8000 m ASL. Following the procedure by e.g. Steen-Larsen et al. (2013), the vapor generation module was used to provide a stream of known and constant water vapor isotopic composition at different humidity steps (from ~500 ppm to ~3500 ppmv). Since the characterization is time consuming, the duration of each step was set to ~15 minutes and only the last 5 minutes of the signal was used to compute the humidity -





isotope response curve. The response curves were repeated for three different laboratory standards (FL0, FL1, FL2; most enriched to most depleted, see Table S1 in supplementary material). For this application, the second MFC and the second mixing tee (MFC2 and TEE2 in Fig. 1) were used while the humidity feedback and the PID control were not used as the calibration system provided a very stable humidity signal to estimate correction curves for the CRDS analyser at very low

humidity, as shown in Fig. 6 Panel a and b. The second mixing tee contributes significantly in producing very small $H_2O$ variability for all steps, yielding a minimum and a maximum standard deviation of 6 and 58 ppmv (22 ppmv, on average). The main disadvantage of the second mixing tee is, however, the use of a large amount of dry air to dilute the signal coming from the oven unit. When using the second mixing tee it is important to pay attention to the internal diameter of the tubing as too large back pressure will inhibit injection of water in the oven through the capillary. For extreme dry conditions (< 300

ppmv) a dedicated low humidity calibration system such as the one proposed in Leroy-Dos Santos et al. (2021) should be adopted.

Interestingly, the CRDS analyser showed slightly different patterns of $\delta^{18}O$ for the different standards used and a very different pattern is observed for $\delta D$ for the very depleted standard as also reported earlier by Weng et al. (2020). Importantly we note that the precision of the CRDS analyser is not affected by the isotopic composition of the standards, as shown in Fig.

6 panel c and d. Users of CRDS analysers at very low humidity must be aware of the different humidity-isotope sensitivity for different isotopic composition, which might be due to issues in fitting of the spectra and could be instrument-dependent.

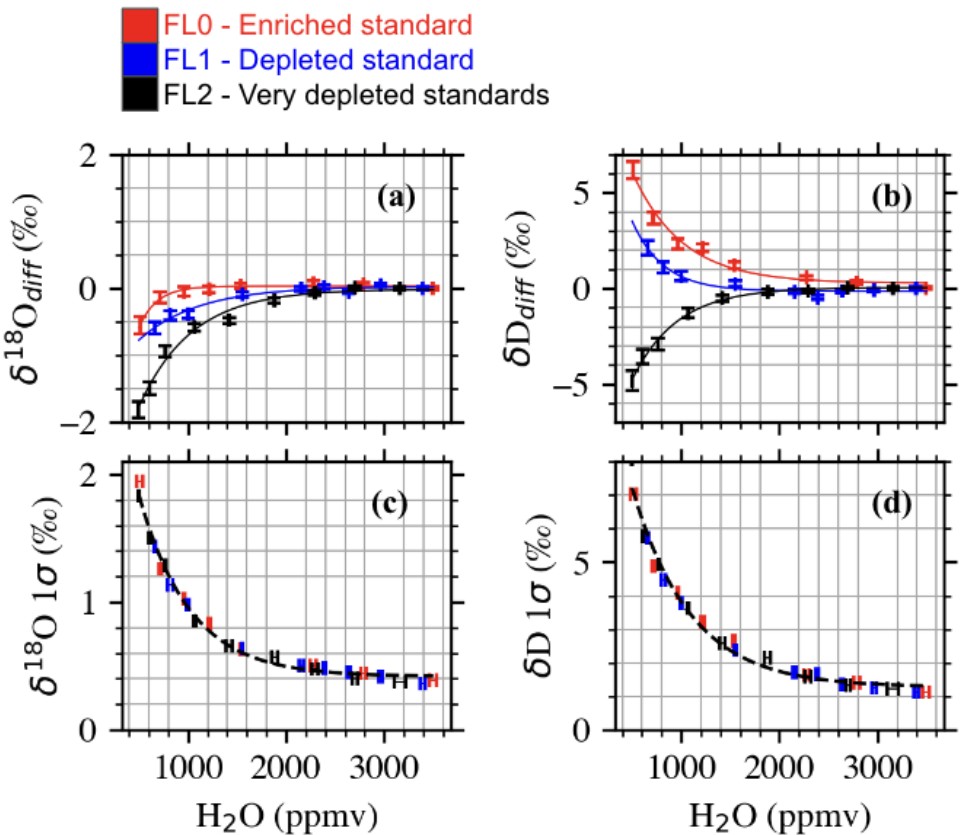

**Figure 6: Humidity-isotope characterization of the HKDS2092 analyzer between 500-3500 ppmv using three different isotopic standards. (a) and (b) humidity - isotope dependency for for $\delta^{18}O$ and $\delta D$, respectively. Dependency reported as the difference between the isotopic composition at 3500 ppmv and isotopic composition for each humidity step. Error bars are standard errors of the mean. Lines are exponential decay models that best fit the observations of each isotope standard. (c) and (d) standard deviation of $\delta^{18}O$ and $\delta D$ for each humidity step and different standards. Error bars on the horizontal axis are standard deviations. For all panels, results of the best fits are reported in supplementary material Table S4.**

## 3.5 Application study 2: $\Delta^{17}O$ analysis of liquid samples

The two reference waters SW and WW, with known $\Delta^{17}O$ values, were mixed in three different aliquots to obtain three solutions with known $\Delta^{17}O$ of 20 ml each (see Table S2 in supplementary material for details on aliquots). The solutions of reference waters were weighted using a laboratory scale with 0.01 g resolution. Half of the maximum span obtained by mixing SW +/- 0.01g with WW -/+ 0.01g yields a 13 per meg variation of the final $\Delta^{17}O$ value, which can be considered a conservative estimate of the sample uncertainty due to weighting. A typical analysis run consisted in injecting the two reference water followed by injections of the two solutions (~15000 ppmv). Each injection lasted for 3 hours and a typical



run consisted of 7 - 12 injections, for a total time of approximately 21 - 36 hours per run. For both reference waters and solutions, the first 2 hours of each injection were discarded to limit the memory effect introduced by the injection of the previous sample. A linear calibration was performed by using reference $\delta^{18}O$ and $\Delta^{17}O$ converted into $\delta^{17}O$. Since the
standards were measured multiple times for each measurement session, and solutions were measured in different days, we applied both an average and independent calibration. The calibration factors for the former were estimated by averaging the results of all the standards of each run. Instead, the calibration factors for the latter were calculated for each couple of reference water and applied to the following couple of solutions (2 standards followed by 2 samples). The results of the experiment are reported in Table 3.


**Table 3: Analysis of $\Delta^{17}O$ of liquid samples using the vapor generation module. All $\Delta^{17}O$ values in per meg. Analytical precision is standard error multiplied by Student's t-factor for a 95% confidence limit. *indicate independent calibrations for each run.**

| Sample | Measured $\Delta^{17}O$ ($\pm 1\sigma$) | Expected $\Delta^{17}O$ | Analytical precision | # replicates (dates Dec 2022) |
|---|---|---|---|---|
| **M20** | 24 ± 3 *24 ± 3 | 21 | 3 *2 | **5** (3 on 9th 1 on 11th 1 on 12th) |
| **M50** | 17 ± 5 *17 ± 4 | 12 | 6 *5 | **4** (2 on 9th 1 on 11th 1 on 12th) |
| **M85** | 24 ± 5 *24 ± 8 | 18 | 5 *8 | **4** (1 on 12th 2 on 13th 1 on 14th) |

Despite the small number of samples, Table 3 clearly shows that reproducibility of sample measurement is very high: 4 and
5 per meg on average for average and independent calibration, respectively. Such reproducibility is comparable to measurement performed with IRMS (e.g. Barkan and Luz, 2005; Steig et al., 2014) and better than analysis performed with optimal settings of vaporizer and autosampler from Picarro, which is 8 per meg following Schauer et al. (2016). The RMSE is 8 per meg and in general the measured $\Delta^{17}O$ is 5-6 per meg higher than the expected one, but within the expected uncertainty due to the weighting. Memory effect was estimated to have a limited impact on $\Delta^{17}O$, as further discussed in
Section 4.4, and we argue the systematic difference can be due to error in the scale, degraded quality of the reference waters due to long storage or biases in assigned $\Delta^{17}O$-values of reference waters. Despite the small bias, this analysis represents a line of evidence that the calibration module can successfully be used to analyse $\Delta^{17}O$ in liquid water given its ability to measure a sample potentially for unlimited time.





## 4 Discussion

### 4.1 Drivers of measurement noise

To investigate the origin of the water isotope measurement noise we placed two Picarro L2140 instruments (HKDS2092 and HKDS2156) in parallel in a configuration that they measured the same output generated by the water vapor generator module at 14500 ppmv level for 10 hours with BER standard. A 1 m length 3.125 mm OD copper tube coming from the water vapor generation module was connected to a stainless steel tee (SS-200-3, Swagelok). Two nearly identical 0.75 m

3.125 mm OD copper tube segments were connected to the tee, carrying the sample gas to each analyser. The instrument connection at the gas inlet was reproduced as similar as possible for both instruments, to achieve the same gas transport resistance. Both instruments were connected to the inlet line in open split mode and each instrument was pulling the gas sample at its nominal flow rate (~40 sccm/min). The water vapor generation module was configured to keep at least a flow rate of 100 sccm/min at its output, to compensate for the requirements of two instruments measuring at the same time.

Exceeding gas sample was released in the room air.

The newer HKDS2156 instrument is performing better than the HKDS2092 in terms of noise and precision, as shown in Fig. 7 panel A. In the figure "bumps" in the Allan Deviation plot for both instruments are seen at different timings for different isotopes and with differences, which are instrument-dependent. This support the hypothesis that the main driver of the noise

in $\delta^{17}O$, $\delta^{18}O$ and $\delta D$ is related to the analysers' characteristics and not to the vapor generation module. A simple correlation analysis of the synchronized and resampled (1 Hz) $H_2O$ signal between the two analysers yielded r = 0.91. Insignificant correlation (<0.01) was observed for $\delta^{18}O$, showing that even though $H_2O$ variability due to vapor generator fluctuations is captured by the two instruments, no co-varying isotopic composition change can be detected by the two instruments during the test (for completeness, $\delta^{17}O$ and $\delta D$ measured with the two analysers have correlation <0.01 and ~0.01, respectively).

Since the variability of the isotopic composition of the signal is independent from the mixing ratio variability, the main sources of isotope variability in the measurement must be investigated in the analyser performances. The isotope data recorded at ~1 Hz by both analysers do not correlate with any of the following parameters recorded in the log file of the analyser: cavity temperature, cavity pressure, DAS temperature, warmbox temperature. However, a wavelet analysis reveals significant power in periods with 10 to 60 minutes periodicity in $\delta^{18}O$ and cavity temperature. When performing a wavelet

coherence analysis (Grinsted et al., 2004) we find a significant in-phase coherence between $\delta^{18}O$ and cavity for periods in the range of 10 to 60 minutes for the HKDS2092 and in the range 10 to 30 minutes for the HKDS2156 instrument (See Appendix Figure B2). When a low-pass Butterworth filter is applied to the data, as shown in a significant correlation between $\delta^{18}O$ and cavity temperature shows (Figure 7 panel B). For instance, a Butterworth low-pass filter with a cut-off of periods less than 600 s yielded r=0.68 for HKDS2092 and r=0.59 for HKDS2156. Correlations also increases for $\delta^{17}O$ and

$\delta D$, but not as much as for $\delta^{18}O$ (e.g., with the same time window, the correlation between $\delta^{17}O$ and cavity temperature is only 0.31 for HKDS2156). It is important to note that the cavity temperature control is precisely tuned using a PID loop and

it is not accessible at user level. We note that a similar "bump" can be seen in Allan Deviation analyses presented in previous studies (e.g. Leroy-Dos Santos et al., 2021; Steig et al., 2014) and we hence speculate if the observed significant periodicity in cavity temperature in the range 12 to 30 minutes is consistent across instruments. Should this hypothesis be correct, it

would imply that $\delta^{18}O$ precision for measurements averaged between $10^2$ and $10^3$ seconds could be improved by up to a factor 2 if the PID driven cavity-temperature cycles were to be dampened.

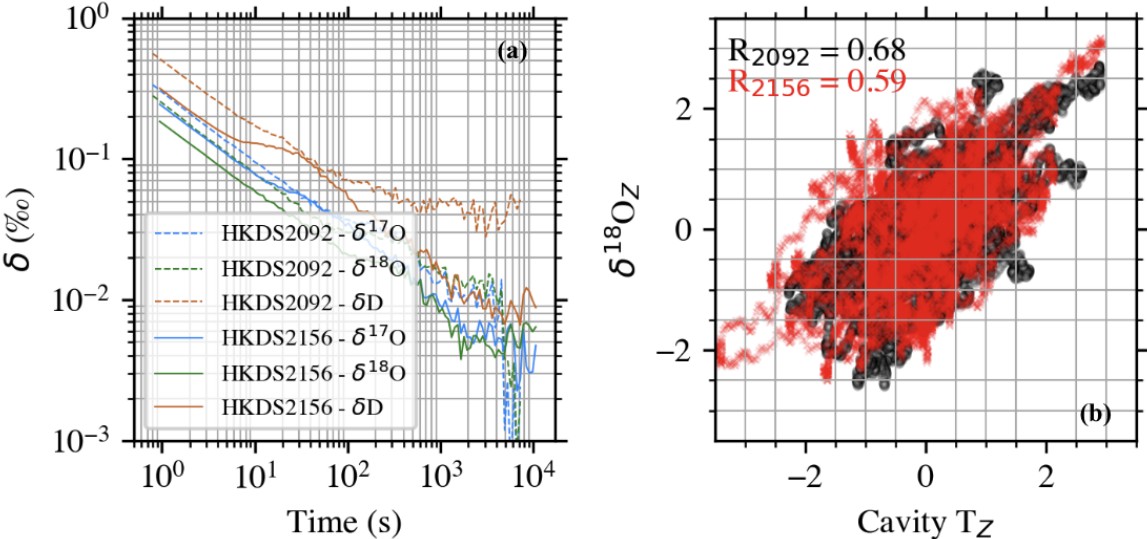

**Figure 7: 10 hours analysis of the same water vapor source using two water isotope analyzers (HKDS2092, HKDS2156). (a) Allan Deviations of the two analyzers showing "bumps" at different averaging time for the two instruments. (b) Z-scores of $\delta^{18}O$ vs z-**
**scores of cavity temperature of the two instruments (HKDS2092 in black, HKDS2156 in red). Standard deviations of $\delta^{18}O$ is 0.02‰ and 0.01‰ for HKDS2092 and HKDS2156, respectively. Standard deviations of cavity temperature is 0.002 and 0.001 K for HKDS2092 and HKDS2156, respectively.**

### 4.2 Isotope calibration pulses

To test the repeatability of the calibration system for producing calibration cycles suitable for water vapor analysis, the
standards BER and SP were used to autonomously produce 100-minute pulse trains spaced every 3 hours for 48 hours continuously. The test was performed with the multiple oven configuration using two different lines to stream from the two different standard 2 ml vials. The mixing ratio level was kept constant at 10 000 ppmv in order to have enough water inside the vials to allow repeated injections for 48 hours. The isotopic composition of the injected water for each pulse, calculated as the average ± 1 standard deviation of the last 5 minutes of the $\delta^{17}O$, $\delta^{18}O$, $\delta D$, d-excess and $\Delta^{17}O$ signals, is reported in
Fig. 8.







**Figure 8: Raw values of repeated injections of BER and SP standards using the multiple ovens configuration. Results reported as average ± standard error calculated for the last 5 minutes of $\delta^{17}O$ (a and b), $\delta^{18}O$ (c and d), $\delta D$ (e and f), d-excess (g and h) and $\Delta^{17}O$ (i and j) signals.**

The calibration system successfully injected both SP and BER standards autonomously, reporting no failed injections for 48 hours. No large change in isotopic composition of the injected standards were observed for 48 hours for the multiple oven configuration. However, as can be observed in Fig. 8 there appears to be slight enrichment in the two standards throughout





the experiment as indicated by the decrease of the d-excess value. This drift of the measurements could be a result of instrumental drift, but we understand the enrichment to be a result of the long exposure of the liquid sample to the dry push gas inside the vial leading to evaporation and hence enrichment. The dry push gas from the head space of the vial being replaced after each injection when stopping the flow of water into the capillary. This observed drift means that whenever a sample in a vial will be measured for a continuous long period of time it is important to use a relative larger sample holder to

limit the relative influence of evaporation. The variability in the mean of the last 5 minutes of the injection is relatively small, with standard errors ranging in the order of 0.01‰, 0.01‰, 0.02‰, 0.1‰ and 11 per meg for $\delta^{17}O$, $\delta^{18}O$, $\delta D$, d-excess and $\Delta^{17}O$ respectively. To illustrate the repeatability of the injections the standard deviation of the mean values from the individual pulses is 0.02‰, 0.02‰, 0.09‰ and 0.13‰ for $\delta^{17}O$, $\delta^{18}O$, $\delta D$, d-excess respectively when linearly correcting for the long-term drift likely induced by evaporation. $\Delta^{17}O$ shows no significant change with time and the

standard deviation of the mean values from the individual pulses is 16 per meg.

This experiment illustrates the potential of the system to be used as a calibration unit for atmospheric water vapor isotope measurements in the field. As we will discuss below the most often occurring issue when using the vapor generation module was clogging of the capillary providing water to the flash evaporator oven. The operator of the system for field calibration of

water vapor isotope measurements therefore needs to use relatively clean standards to allow for extended operation. As illustrated in the long-term stability experiment in Fig. 3 we have carried out a successful ~90 hour long injection of a single standard. While we have not had the opportunity to test this in practice, the system could in principle provide a more than 3 month long autonomous calibration when measuring a standard for 1 hour per day.

**4.3 Capillary issues**

To control the flow of water into the oven as precisely as possible it is beneficial to use a capillary with a small internal diameter between 50um and 350um. However, as a consequence of the flash evaporation at the tip of the capillary inside the oven residue from dissolved and non-dissolved impurities in the water is building up ultimately leading to clogging of the capillary. Typical signs of clogging are gradual decrease of humidity and enhanced variability in the humidity. Following Gkinis et al. (2011) we started out using a silica capillary but discovered that using a stainless steel capillary (T10C5, VICI)

resulted in enhanced performance in terms of length of operation before signs of clogging appeared. We also observed that using a stainless steel capillary had the advantage that it could be placed the same way every time, while due to the flexibility of the capillary it occurred that the capillary would touch the nut of the oven leading to a non-uniform heating at the tip. Such cases of non-uniform heating were often seen as increased variability in the generated humidity. We do not have an explanation for why the stainless steel capillary was performing better. From experience we observed that using a

capillary with an ID of 127 um created the overall best performance. Once a capillary was clogged, we had partly success to unclog it by placing it in an ultrasonic bath with deionized water. As we will discuss in details below the clogging of the capillary had the consequence of changing the memory effect.



## 4.4 Memory effect

As documented in Section 3.2 memory effects need to be considered, when using the vapor generation module for
measurements of samples with a large spread in water isotopic composition. The vapor generation module shows an approximate factor four reduction in memory effect for $\delta D$ compared to liquid injections using an autosampler and vaporizer. However, in the work-mode applied here, the control software will automatically adjust the dry air dilution to maintain a targeted humidity setpoint. The consequence of this is that during a build-up of non-dissolved impurities less water will be delivered at the tip of the capillary, and the control software will reduce the flow of dry air. Less water and air
molecules will therefore flow through the system leading to a reduction in removal of water molecules from the previous sample. An example of such situation is illustrated in supplementary material Fig. S2 for a similar pulse train of standard injections as discussed in Section 4.2 with approximate jump of 55‰/435‰ in $\delta^{18}O/\delta D$. We do not know of similar published tests with other types of vapor generation modules, but we expect that similar variability in memory effect must also be present for other modules. Our presented example therefore stands as a lesson in the need for quantifying and
tracking the performance of every aspect when measuring standards and unknown samples. When using the vapor generation module in automatic measurement mode, one must therefore characterize the memory effect of the setup as a function of dilution flow rate and have the control system log the flow rate for post memory correction of the sample and standard measurement.

The difference in memory effect for the individual isotopes as documented in Fig. 4 illustrates the need to reject relatively longer measurement-time in order to get accurate $\delta D$ measurements compared to $\delta^{18}O$. We speculate that the difference in memory effect between $\delta D$ and $\delta^{18}O$ originates due to the difference in polarity of the $HD^{16}O$ and $H_2^{18}O$ molecules with the $HD^{16}O$ adhering relatively stronger to the sides of the tubing. This difference in memory effect of $\delta^{18}O$ compared to $\delta D$ is not only important for obtaining the most accurate isotope values, but also means that care needs to be taken when
calculating the d-excess. Interesting, the memory effect for $\delta^{18}O$ and $\delta^{17}O$ is nearly identical. Due to the relatively high measurement uncertainty and relatively small changes in $\Delta^{17}O$, it is not the memory effect, which is the limiting factor influencing the $\Delta^{17}O$ measurements. As documented in Fig. 9 showing the memory effect for a $\Delta^{17}O$ step change of approximately 90 per meg, already within the first hour of measurements the memory effect is producing an error, which is ~five times smaller than the analytical error. Hence, the vapor generation module is optimal for $\Delta^{17}O$ as it can provide in
principle unlimited integration time to reduce the noise.





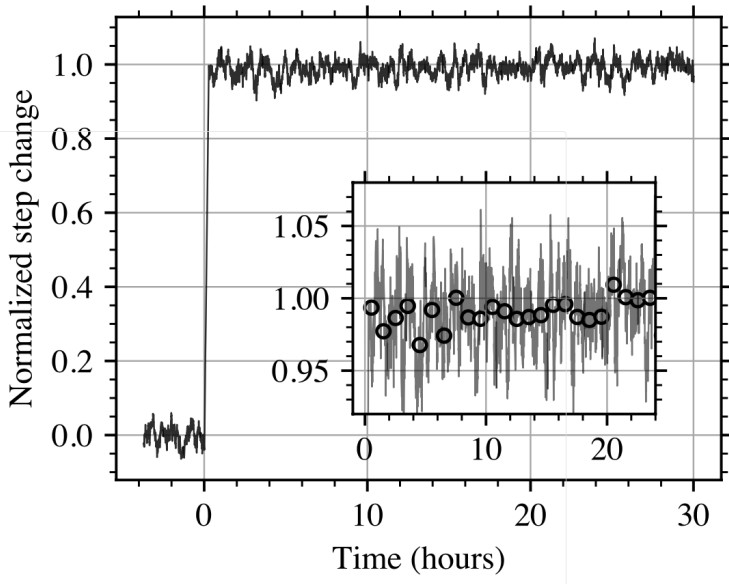

**Figure 9: Normalized step change for $\Delta^{17}$O. Black line is a 15-minute window moving average. Inset plot: detail of the 0-24h region. Grey line is the same as in the main plot. Circles are 1-hour averaged sample mean.**

### 4.5 Potential future updates

We developed vapor generation module system with 4 ovens allowing 4 individual vials to be measured without interference from an operator. However, by connecting one of the ovens to a 10-port selector valve we were able to run the system for liquid measurements using 3 known standards and 10 unknown samples. We are now in the process of setting up the system for a standard daily operating procedure measuring $\Delta^{17}$O of ice core samples. We have nevertheless observed that as a consequence of using the selector valve the capillary connected to the selector valve will clog up more frequently as

relatively more liquid per day is pushed through it. As part of the development of the standard daily operating procedure we will need to develop a protocol for cleaning the capillary of debris or use a larger inner diameter. To increase further throughput by reducing the memory effect, one could also configure the N-oven system to initiate the vapor generation of the subsequent vial while the previous vial is being measured. It would then be a matter of simply switching to an already existing vapor stream from the next oven in line. We invite readers, who plan to build a similar module to contact us for the

latest update on measurement procedure and control software.

### 5 Conclusion

The water vapor generation module presented here represents a stand-alone modular system for water vapor isotope calibrations in the field and high precision liquid water isotope measurements in the laboratory. The system is completely scalable from one sample/standard vial up to N vials. The vapor generation module is designed to be connected directly with



a water vapor isotope analyser without the need for additional auxiliary systems. There are several advantages of the system compared to existing systems for vapor calibration and liquid sample measurements for water stable isotopes, most notable the possibility of calibrating using several standards, the high dynamic range of humidity generation, significant reduction in memory effect, and the possibility to, in principle, measure indefinitely the same sample.

We have used the vapor generation module to characterize the performance of two water isotope analysers, and documented

that a key driver of the measurement noise for averaging times between 10 minutes and 1 hour is driven by temperature fluctuations of the cavity. This result provides guidance on optimal measurement integration times as well as points to the need for stabilizing the cavity temperature on these time scales. We argue that previously published increases in Allan Deviation for longer averaging times could be a result of memory effects and not only driven by instrumental drifts as it is often interpreted as. We document Allan Deviation of the water vapor isotope analyser yielding 0.004, 0.006, 0.01, 0.03 ‰

$\sigma_{\text{Allan}}$ for $10^4$ seconds averaging time for $\delta^{17}O$, $\delta^{18}O$, $\delta D$, d-excess respectively and 2 per meg $\sigma_{\text{Allan}}$ for $10^4$ seconds averaging time for $\Delta^{17}O$.

Despite the memory effect of the vapor generation module being respectively 2 and 3 times smaller compared to injections using an autosampler for $\delta 18O/\delta 17O$ and $\delta D$, it is still possible to detect the memory effect after more than ~20 hours, thanks to the high measurement precision on the instrument. This result indicates that for achieving measurements with very

low uncertainty as for example needed for ice core analysis of million-year-old ice, dedicated efforts on developing measurements systems with minimal memory effect is needed. We find however, that measurements of $\Delta^{17}O$ is limited by the integration time of the vapor measurements and to a lesser extend the memory effect. By measuring an unknown liquid sample for $\Delta^{17}O$ for a three-hour period (rejecting the first two hours due to memory effect) we document an average standard deviation of 4 per meg and an average standard error (95 % confidence limit) using 4-5 replicates of 5 per meg.

It has been speculated that the measurement uncertainty using laser spectroscopy instruments would increase for more depleted values. Using standards spanning of 55‰/435‰ in $\delta 18O/\delta D$ we do not find any evidence that measuring more depleted values results in higher uncertainty.

Using sufficiently clean water we have been able to continuously measure a constant vapor stream from a single standard over a period of more than 90 hours. This documents the feasibility of using the vapor generation module as an autonomous

field calibration system. The PID control system ensuring stable humidity levels along with the large range of possible humidity levels make the vapor generation module truly versatile both in the field and in the laboratory.






# 6 Appendices

**Appendix A: Stability at different humidity levels**

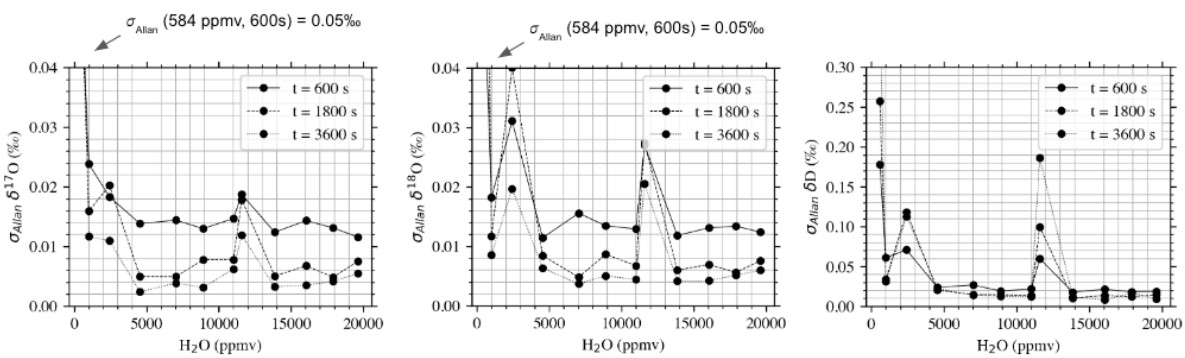

**Figure A1: Allan Deviation for $\delta^{17}O$, $\delta^{18}O$ and $\delta D$ at 600, 1800, and 3500 second integration time as function of humidity level.**

**Appendix B: Squared Wavelet Coherence analysis between $\delta^{18}O$ and cavity temperature**

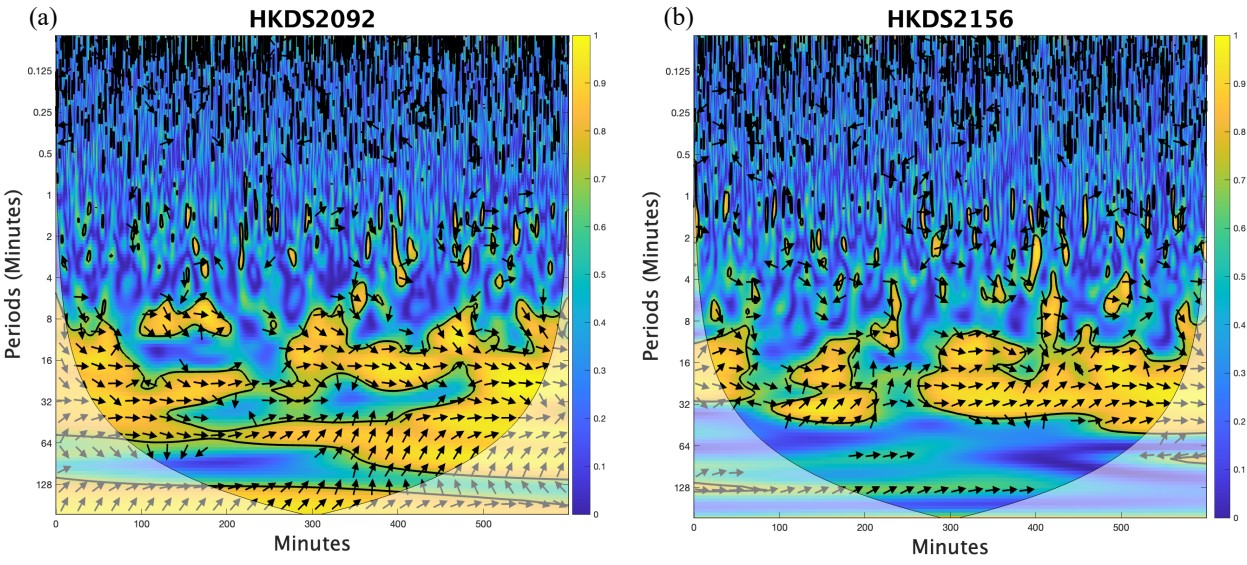


**Figure B1: Squared Wavelet Coherence analysis between the measured $\delta^{18}O$ and recorded cavity temperature of the two Picarro analyzers HKDS2091 (Panel a) and HKDS2156 (Panel b) using procedure by Grinsted et al. (2004). The 5% significance level against red noise is shown as a thick contour. Arrows indicate phasing (angle corresponds to phase behaviour). In-phase behaviour of $\delta^{18}O$ and recorded cavity temperature is observed for periods between ~12 minutes and ~30 to 60 minutes for respectively**
**HKDS2156 and HKDS2092 analysers.**



**Author Contributions**

HCSL conceptualizes this study. HCSL and DZ carried out the development of the methodology and investigation. DZ led with HCSL the formal analysis. DZ carried out the visualization. DZ carried out data curation. HCSL and DZ wrote the original draft, edited, and reviewed the submitted version. HCSL acquired funding for this study and administrated the project.

**Competing Interests**

The authors declare that they have no conflict of interest.

**Acknowledgements**

This work has received funding from the European Research Council (ERC) under the European Union's Horizon 2020 research and innovation program: Starting Grant SNOWISO (Grant Agreement no. 759526).

We are grateful to Laboratoire des Sciences du Climat et de l'Environnement, Centre for Ice and Climate at the Niels Bohr Institute, and the Stable Isotope Laboratory at the Institute of Arctic and Alpine Research, University of Colorado for providing samples of their internal standards for use in this study. The 17O-excess values of the standards provided by the Stable Isotope Laboratory were provided by Δ*IsoLab, University of Washington. DZ acknowledges funding from the European Union's Horizon 2020 Research and Innovation program under Grant 821868 and from the European Space Agency under EO science for society permanently open call Grant 4000134119/21/I-DT-lr. We are grateful for the many who have provided input to earlier versions of the prototype of the vapor generation module used during SNOWISO field campaigns in Greenland. We thank Picarro Inc. for the instrument support making the instrument comparison analysis possible.

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
