# Peer review of "A versatile water vapor generation module for vapor isotope calibration and liquid isotope measurements"

_Atmospheric Measurement Techniques, 2023_

## Referee Comment (RC1)

Review of Steen-Larsen et al AMT manuscript.

1. Does the paper address relevant scientific questions within the scope of AMT? YES
2. Does the paper present novel concepts, ideas, tools, or data? YES
3. Are substantial conclusions reached? YES
4. Are the scientific methods and assumptions valid and clearly outlined? Good, but can be improved.
5. Are the results sufficient to support the interpretations and conclusions? YES
6. Is the description of experiments and calculations sufficiently complete and precise to allow their reproduction by fellow scientists (traceability of results)? Generally yes, but could be more clear
7. Do the authors give proper credit to related work and clearly indicate their own new/original contribution? YES
8. Does the title clearly reflect the contents of the paper? YES
9. Does the abstract provide a concise and complete summary? YES
10. Is the overall presentation well structured and clear? YES
11. Is the language fluent and precise? YES
12. Are mathematical formulae, symbols, abbreviations, and units correctly defined and used? YES
13. Should any parts of the paper (text, formulae, figures, tables) be clarified, reduced, combined, or eliminated? YES- See comments
14. Are the number and quality of references appropriate? YES
15. Is the amount and quality of supplementary material appropriate? YES

**General Comment:**

This manuscript presents a comprehensive and detailed explanation of methods and apparatus that keenly solve some of the formidable challenges that have perplexed field scientists attempting to make and calibrate high quality measurements of water isotopes and water vapor concentration for some time. For those who can successfully implement the technical designs and concepts presented here and apply it to their research, I am convinced it will represent a step change in their ability to make more useful and reliable measurements in a variety of environments. It is clear that this represents a culmination and evolution of concerted efforts, both on the part of the authors, and many whom they cite, in a long term quest to address the challenge of stability in analytical systems of this type. I whole heartedly recommend its acceptance for publication, with some minor edits, mostly technical clarity, and slight improvements to 2 of the figures. If possible, a few photos would really help illuminate the not only the complexity of the system, but it's likely compact nature and portability.

One of the more important innovations is the advanced PID control of the headspace pressure in the vials by metering between pressurized air and vacuum, leading to remarkable steady metrics in both isotopes and water vapor concentration. An equally head-line worthy finding is

the speculation on the effects that cavity temperature control have on the precision of the Picarro instrument. This may not be surprising to some but seeing it here along with all the other metrics is compelling. Not only have the authors achieved in creating a remarkable instrument/inlet system, but they have tested the system thoroughly and produced metrics that quantify and demonstrate its stability in a convincing manner.

**Detailed Line by Line:**

Line 23: Typo: "… as a calibration system have been document to …" -> should be "as a calibration system has been documented".

Line 25: Add ", assuming 1 hour/day for calibration" to the end of the sentence.

Line 42: Add parenthesis to the d-excess equation: *d-excess* = $\delta$D - (8 x $\delta^{18}$O)

Line 77: The use of a selector valve is mentioned, but is not shown in figure 2.

Line 180- 190: Mixed past and present tense (was and is).

Line 186: An explanation of why a PTFE fitting is used instead of stainless steel would help.

Line 196: The use of the term "oven" is a bit confusing or misleading. Consider using Heated assembly, or perhaps heated tees, or reaction vessels, or? The word oven connotes and enclosure of some kind, which is not really the case?

Line 202: "The water vapor is produced in the oven is routed toward the output of the system…." How? Stainless steel capillary or?. Specify here.

Line 208: Spacing issue in the word "different".

Line 209: "In a second modification…" Was there a first modification? Perhaps it could just be 'In a subsequent modification' or similar.

Line 233: Spacing issue in the word "analyzer".

Line 275: It is not clear what is meant by "…Suboptimal management of memory effect[s] during Allan Deviation tests." Is this in the Picarro instrument, or in the inlet system described here?

Line 289:        The authors system really deserves a fancy acronym name of some type 😉 as opposed to "the vapor generation module".  Not just for cache, but to more easily distinguish it from the "vaporizer using the autosampler".   Perhaps one could refer to the Picarro vaporizer/autosampler combination, or similar?

Line 291:        Consider adding something like: " The large difference in these values is discussed further in section 4.4 on memory".

Line 320-324:  This stable region at ~12,500 is a very interesting finding.  Any speculation as to what may govern this (?) could be useful.

Line 342:        Perhaps also include Rozmiarek et al. (2021).  [https://doi.org/10.5194/amt-14-7045-2021].

Line 408:        "A  1 m length of 3.125 mm OD copper tube".  Maybe explain why such a long length was needed?

Line 419:        Typo "This support[s] the hypothesis…"

Line 430:        Typo (?)  "…in phase coherence between $\delta^{18}O$ and cavity [?] for periods…"  I think you mean cavity temperature and/or pressure?

Line 439-441:  This is a MAJOR finding that the precision for measurements could be improved by up to a factor of 2 if the PID driven cavity-temperature cycles could be dampened.   Might deserve higher placement or highlighting some how.  No doubt, manufactures will be very interested in this, along with the many other findings !

Line 493-494:  The reader will greatly appreciate the honesty implicit in the statement "We do not have an explanation for why the stainless steel capillary was performing better"!

Line 495:        "…we had partly success"  should be 'we had partial success'.

Line 496:        "As we will discuss *in* details below[,]"  Consider:  'As we will discuss in detail below[,] the clogging…etc.'

**Figures and Tables:**

Figure 1:  This needs some work.  While schematically correct (after considerable time discerning this), it could be made far more accessible to the general reader, by including some more details and labels.  For example,

- label (or describe in the caption) the type of compressed air (nitrogen, zero air, or?);

- labeling the 3-way metering valves as such (these are key to the design);
- Instead of the part number of the R/H probe, consider putting that information in the text or caption, and simply call it for what it is "Precision R/H probe or similar".
- How do 4 "ovens" converge into a single outlet?
- Is there a selection valve missing from the diagram?
- Oven unit could use a foot note "See figure 2" in its label
- If AMT allows use of color(s), consider using them to further distinguish flows of air, water and water vapor, with thicker lines.  Or use words on the lines.
- Label the 1-4 psi pressure regulator (?) as such.
- Use bigger font were possible
- Not clear what the dashed red lines are, other than T1 and Tn, indicating there could be multiple vials.
- Is the input to MFC 1 really an unregulated line to the compressed gas?  I suspect a regulator is in there somewhere (in addition to the 1-4 psi).
- Output could include "open split to analyzer" or such.

Figure 2:   Drawing of the heated tee sections is good.  Some improvements could include:

- Consider adding the word  "oven[s]" to the block of heated tees.  (I know its in the caption, but will help the reader).
- Consider showing some type of dry air inputs on the left, $H_2O$ on top, and vapor coming out the right side more clearly .  Maybe add some capillaries?
- Arrow pointing to PTFE capillary is actually pointing to he Swaglok nut (picky, I know).
- Consider labeling the bottom of the assembly with "2 ml vials with water" or such.

Figure 3, 4, 5, 6, 7 & 9: All Good

Figure 8:  Cosmetic improvement: Consider smaller fonts.

Tables

Table 1:  Good. It may be obvious, but consider spelling out in the caption that BER is  Bergen water and SP is South Pole water.

Table 2:  Good.  Need to widen $2^{nd}$ column a tiny bit to better accommodate heading text.  Consider defining RSD either here in the caption or in the text.

Table 3:  Good.

---

## Author Comment (AC1)

**Response to Reviewer 1 on review of "A versatile water vapor generation module for vapor isotope calibration and liquid isotope measurements"**

We thank Reviewer 1 for the detailed comments to our manuscript and for taking the time to provide constructive feedback.
We have below responded to the individual comments of the reviewer and indicated how we have improved the manuscript. We have used RED font to indicate our response. We appreciate the reviewer´s comments on improving the figures. In general we have followed all of the suggestions by the reviewer.

**Reviewer 1**
This manuscript presents a comprehensive and detailed explanation of methods and apparatus that keenly solve some of the formidable challenges that have perplexed field scientists attempting to make and calibrate high quality measurements of water isotopes and water vapor concentration for some time. For those who can successfully implement the technical designs and concepts presented here and apply it to their research, I am convinced it will represent a step change in their ability to make more useful and reliable measurements in a variety of environments. It is clear that this represents a culmination and evolution of concerted efforts, both on the part of the authors, and many whom they cite, in a long term quest to address the challenge of stability in analytical systems of this type. I whole heartedly recommend its acceptance for publication, with some minor edits, mostly technical clarity, and slight improvements to 2 of the figures. If possible, a few photos would really help illuminate the not only the complexity of the system, but it's likely compact nature and portability.

One of the more important innovations is the advanced PID control of the headspace pressure in the vials by metering between pressurized air and vacuum, leading to remarkable steady metrics in both isotopes and water vapor concentration. An equally head-line worthy finding is the speculation on the effects that cavity temperature control have on the precision of the Picarro instrument. This may not be surprising to some but seeing it here along with all the other metrics is compelling. Not only have the authors achieved in creating a remarkable instrument/inlet system, but they have tested the system thoroughly and produced metrics that quantify and demonstrate its stability in a convincing manner.

Detailed Line by Line:
Line 23: Typo: "… as a calibration system have been document to …" -> should be "as a calibration system has been documented".
Corrected.

Line 25: Add ", assuming 1 hour/day for calibration" to the end of the sentence.
Corrected

Line 42: Add parenthesis to the d-excess equation: d-excess = dD - (8 x d18O)
Corrected

Line 77: The use of a selector valve is mentioned, but is not shown in figure 2.

New schematic in manuscript. Schematic with selector in supplementary material.

Line 180- 190: Mixed past and present tense (was and is).
Corrected

Line 186: An explanation of why a PTFE fitting is used instead of stainless steel would help.
PTFE ferrule and nut have been used to not deform the PTFE capillary. Text inserted.

Line 196: The use of the term "oven" is a bit confusing or misleading. Consider using Heated assembly, or perhaps heated tees, or reaction vessels, or? The word oven connotes and enclosure of some kind, which is not really the case?
We here use the term oven to represent both the heating block and the heated tees. We have however specified the text such that it refers to "module is highly scalable in terms of number of heated tee units" instead of oven units.

Line 202: "The water vapor is produced in the oven is routed toward the output of the system…." How? Stainless steel capillary or?. Specify here.
We have updated the text such that it now reads:
…is routed toward the output of the system through the stainless steel valve manifold and a copper tube where the vapor is measured using…

Line 208: Spacing issue in the word "different".

Line 209: "In a second modification…" Was there a first modification? Perhaps it could just be 'In a subsequent modification' or similar.
We will use the term "subsequent" - this refers to the use of a selector valve "multiport selector (C25-3180EUHA, VICI) to inject samples from the different vial holders continuously, without needing to switch the oven" instead of an multi-oven unit.

Line 233: Spacing issue in the word "analyzer".

Line 275: It is not clear what is meant by "…Suboptimal management of memory effect[s] during Allan Deviation tests." Is this in the Picarro instrument, or in the inlet system described Here?
We are here referring to the memory effect due to the combination of calibration system, the inlet and Picarro's cavity. We don't know if authors of the mentioned studies had let the system to "prime" for 12 hours or so and hence we have introduced the sentence ", such as not priming the system for upto 12 hours using the same standard".

Line 289: The authors system really deserves a fancy acronym name of some type 😉 as opposed to "the vapor generation module". Not just for cache, but to more easily distinguish it from the "vaporizer using the autosampler". Perhaps one could refer to the Picarro vaporizer/autosampler combination, or similar?
Thank you for this suggestion - we have proposed to name it VG Module

Line 291: Consider adding something like: " The large difference in these values is discussed further in section 4.4 on memory".
Good idea - corrected.

Line 320-324: This stable region at ~12,500 is a very interesting finding. Any speculation as to what may govern this (?) could be useful.
We believe that it could be the point of "stable equilibrium" between pressure-driven liquid flow rate and the dry air flow of the calibration system?

Line 342: Perhaps also include Rozmiarek et al. (2021). [https://doi.org/10.5194/amt-14-7045-2021].
Added
Line 408: "A 1 m length of 3.125 mm OD copper tube". Maybe explain why such a long length was needed?
The length of the tubing could have been reduced if placed on the same table top. However, the lengths were adjusted to enable connections of the instruments placed over different tables. Clearly to improve the memory effect further one should decrease the length of tubing.

Line 419: Typo "This support[s] the hypothesis…"
Corrected

Line 430: Typo (?) "…in phase coherence between d18O and cavity [?] for periods…" I think you mean cavity temperature and/or pressure?
Correct - we meant cavity temperature
Line 439-441: This is a MAJOR finding that the precision for measurements could be improved by up to a factor of 2 if the PID driven cavity-temperature cycles could be dampened. Might deserve higher placement or highlighting some how. No doubt, manufactures will be very interested in this, along with the many other findings !
Thank you. We further notice that thes "bump" is occurring for the same integration time as liquid sample measurements. We have further expanded on this in the abstract and added the following sentence ", which if improved upon could result in an improvement in measurement precision of up to a factor 2."

Line 493-494: The reader will greatly appreciate the honesty implicit in the statement "We do not have an explanation for why the stainless steel capillary was performing better"!
Thank you. We do try to also highlight that we not always know why it works better.

Line 495: "…we had partly success" should be 'we had partial success'.
Corrected

Line 496: "As we will discuss in details below[,]" Consider: 'As we will discuss in detail below[,] the clogging…etc.'
Corrected

Figures and Tables:

Figure 1: This needs some work. While schematically correct (after considerable time discerning this), it could be made far more accessible to the general reader, by including some more details and labels. For example,

• label (or describe in the caption) the type of compressed air (nitrogen, zero air, or?);• labeling the 3-way metering valves as such (these are key to the design);

• Instead of the part number of the R/H probe, consider putting that information in the text or caption, and simply call it for what it is "Precision R/H probe or similar".

• How do 4 "ovens" converge into a single outlet?

• Is there a selection valve missing from the diagram?

• Oven unit could use a foot note "See figure 2" in its label

• If AMT allows use of color(s), consider using them to further distinguish flows of air, water and water vapor, with thicker lines. Or use words on the lines.

• Label the 1-4 psi pressure regulator (?) as such.

• Use bigger font were possible

• Not clear what the dashed red lines are, other than T1 and Tn, indicating there could be multiple vials.

• Is the input to MFC 1 really an unregulated line to the compressed gas? I suspect a regulator is in there somewhere (in addition to the 1-4 psi).

• Output could include "open split to analyzer" or such.

Figure 1 edited following reviewer comments.

Figure 2: Drawing of the heated tee sections is good. Some improvements could include:

• Consider adding the word "oven[s]" to the block of heated tees. (I know its in the caption, but will help the reader).

• Consider showing some type of dry air inputs on the left, H2O on top, and vapor coming out the right side more clearly . Maybe add some capillaries?

• Arrow pointing to PTFE capillary is actually pointing to he Swaglok nut (picky, I know).

• Consider labeling the bottom of the assembly with "2 ml vials with water" or such.

Figure 2 edited following reviewer comments.

Figure 3, 4, 5, 6, 7 & 9: All Good

Figure 8: Cosmetic improvement: Consider smaller fonts.

Tables

Table 1: Good. It may be obvious, but consider spelling out in the caption that BER is Bergen water and SP is South Pole water.

In fact BER is Bermuda water 🙂

Updated

Table 2: Good. Need to widen 2nd column a tiny bit to better accommodate heading text. Consider defining RSD either here in the caption or in the text.

Corrected

Table 3: Good.

---

## Author Comment (AC2)

**Response to Reviewer 2 on review of "A versatile water vapor generation module for vapor isotope calibration and liquid isotope measurements"**

We thank the reviewer for the time commitment to carry out the review of our manuscript and for providing insightful questions, which have allowed us to discuss further the performance of the vapor generation module. We have provided answers to the comments using red font below.

General comment

This paper presents a new water vapor generation module meant to improve the accuracy and precision of the analysis of water isotopes in the liquid or vapour form, which is a critical issue for the study of certain second-order parameters like d-excess or 17O-excess in atmospheric water vapour or ice cores. The presented calibration module is an efficient technical solution to solve a number of problems encountered by the authors and listed in the introduction, namely reduce the memory effect, increase the robustness and reliability for field calibration and adapt the system for multiple standard analysis while offering a large humidity range and a stability over several days. These technical issues are indeed encountered by many teams working on this specific subject and the first part of the articles gives a very detailed description to duplicate the proposed solution, including technical references.

In a second part, the performance of a Picarro analyser and the vapor generator are presented and discussed, using several tools such as the Allan deviation or the wavelet coherence analysis. By comparing two Picarros, or comparing the new vapor generator module and a commercial Picarro autosampler and vaporizer, the authors are able to determine whether the performance originate from the analyser or the vapor generator module in a very convincing way.

Scientific questions

I have a few questions and comments that might shed some additional light on your discussions:

You demonstrated in this paper a reduction of the memory effect using the new vapor generator module compared to the Picarro vaporizer, especially visible on dD. **Can you precise whether the residual memory effect is dominated by the Picarro response or the humidity generator?**
We believe the residual memory effect to be dominated by the Picarro response, specifically by the flushing time and the volume of the measurement cavity. The calibration system has smaller dead volumes (and specific surface area) on the stages following vaporization than the measurement cavity itself. Accounting for the oven, the outlet manifold, the outlet line and the humidity sensor, the estimated volume is 10 - 11 cm3, roughly ⅓ of the Picarro's cavity. Moreover, the calibration system is flushed usually above 50 sccm/min (usual in the range of

200 sccm/min depending on targeted humidity level), which is higher than the Picarro's flow rate.

The wavelet coherence analysis gives a good indication on the correlation between the cavity temperature and the delta measurement. **Did you plot the Allan deviation of some of the studied Picarro parameters (cavity temperature, pressure, etc) to check for the presence of the same bump?** It could be interesting to compare it on the two Picarro analysers who do not show the same bump. Also, other parameters such as the cavity temperature or warm box PWM can be interesting to check.

To further shed light on our finding we provide additional figures below, which has not been included in the manuscript. First we report the Allan deviation for Cavity Temperature and Cavity pressure of the two instruments (HKDS2092 dashed lines, HKDS2156 solid lines). The two analyzers have very similar Allan deviation shape, but no evident feature at the same timing of the bump in the delta measurement. The bump in the delta measurement might be due to non-stationary effects in the spectroscopic measurement system, since they occur at different timings between the two different instruments.

We further show low pass filtered d18O of the two instruments on the common time scale to illustrate that the increased variability in d18O at these integration times are not common between the two instruments.

Further we show the low pass filter of the cavity temperature and the low pass filter of the d18O, which ultimately is the values that goes into Figure 7. However, here we show the variability on a time scale, which indicate very strong coherency.

We also show the wavelet cohency between the DAS temperature and the cavity temperature to understand if the variability in the cavity temperature were forced by the room temperature variation. This hypothesis we reject, which leads us to conclude that it is likely an internal variability in the cavity temperature that is at play.

[Figure]

[Figure]

Technical corrections

The legend of fig. 7 should be removed or made smaller to avoid covering the Allan deviation curves
Figure 7 edited following reviewer comments.

Table 2 could be easier to read in a graphic way, by putting for example the humidity as x-axis and the other metrics as y-axis
We follow the advice of the reviewer and make a new figure to illustrate tabel 2. We also place Table 2 in supplement material.

For an easier reading of fig. 8, I suggest a centering of the delta values around zero (by subtracting the mean value to the raw dataset) and share the same y axis for d17O (a and b), d18O (c and d), dD (e and f), dexcess (g and h) and D17O (i and j).
Figure 8 edited following reviewer(s) comments.

Fig. 3: If possible, I would be interested in seeing the temporal signal of d18O and dD (below the H2O curve for example). Maybe with a rolling average the memory effect can be directly observed?
Attached below is the time series (rolling average window 1800 s). We have also included this in the supplement material.

[Figure]

Fig. 2: I would be interested in having a global 3D drawing of the water vapor generator to understand how the two pieces are connected. Otherwise, a photo of the module would be appreciated.

Photos are attached to the supplementary material.

Citation: https://doi.org/10.5194/amt-2023-160-RC2

---

## Author Comment (AC3)

**Response to Reviewer 3 on review of "A versatile water vapor generation module for vapor isotope calibration and liquid isotope measurements"**

*We thank Reviewer 3 for the detailed comments. While we have not hastily put together the manuscript, we have made note of the comment by the reviewer on the readability of the manuscript and have therefore worked towards improving the language. We have addressed the comments by the reviewer below in* red font.

The manuscript "A versatile water vapor generation module for vapor isotope calibration and liquid isotope measurements" by HC Steen-Larsen and D Zannoni describes an excellent addition for those in the laser spectroscopy community interested in pushing the limits of their water isotope instrumentation. It clearly meets the scope of Atmospheric Measurement Techniques and addresses a frustration by many atmospheric water vapor researchers by providing a method for reliable and automatic calibration of their vapor measurements across a wide range of concentrations. The manuscript should be accepted for publication in AMT after attending to my comments and suggestions described below.

My main criticism is that the manuscript frequently reads like a rough draft that was hastily put together and submitted. To be clear, the content of the study is sound, the science is sound, all of the pieces are present, and it is encouraging that my only criticism is related to the conveyance of the material. Still, the authors need to read through it very carefully, ask a colleague to proofread it, or preferably, both. While the team at AMT will help provide a final polished version, the authors need to take it upon themselves to do much of the work.

Having gone through the manuscript with a careful eye we agree with the reviewer that too many grammatical errors were present and have (hopefully) improved the manuscript.

The measured vs expected results presented in Table 3 are clouded by a confusing description of how these data were calibrated. **I would not be able to reproduce your calibration strategies based on lines 384 to 389. Are the mixtures calibrated to the pure versions of SW and WW or to SP and BER?**
Yes, correct, we use the SW and WW to calibrate the mixtures in the run. The text in the manuscript is now updated.

A typical measurement session (run) is composed of analysis of two standards followed by analysis of two samples, as shown below. Independent calibration means that each run is calibrated using the two standards of that specific run. Average calibration means that the results of the analysis of the standards have been averaged and an average calibration line was built to calibrate the analysis of the samples.

[Figure]

This procedure is described in the text as follows:

*Since the standards were from 2 to 4 times for each measurement session, and the solutions were measured on different days, we applied both an average and independent calibration. The average calibration factors were estimated by averaging the raw observations of the standards injected for all the run. Instead, the independent calibration factors were calculated for each couple of reference standards and then applied only to the following couple of solutions (2 standards followed by 2 samples). The results of the experiment are reported in Table 3.*

While within error, the bias is interesting and points to a calibration issue. Do the authors have calibration weights for their balance used to create the mixtures? The authors provide no evidence to support the claims in the sentence spanning line 400.
These are some of the most likely issues affecting our results, hence, there is no way to prove them.

What is the maximum consumption rate of liquid water from a 2 mL vial?
We estimated the amount of water required by the system via mass balance (known dry air flow rate between 50 and 500 sccm/min, 100% vaporization, no leaks, calibrated $H_2O$ Picarro readings). **The actual consumption rate of liquid is estimated to be ~1.5 µl/min,** which dries out a **2ml vial in ~22 hours**. This is consistent with our observations/experience.
We have added this information in section 3.1.

Given the variety of concentrations one could choose from for a deployment, what volume of reservoir do the authors recommend, or perhaps more to the point of this paper, what is the maximum reservoir volume given the dry-air-pump enrichment noted on line 465.
We would recommend a volume between 2 and 10 ml, and possibly to not dry out the vial in order to minimize the volume of water exposed to dry air in the head space (i.e. large volume of water, small volume of push gas/air). 2 ml of standards (inside a 10 ml vial) should be enough per one calibration hour for nearly three weeks, likely more, and to minimize the dry-air-pump enrichment <<0.05‰ for d18O, between start and end.

Regarding the data presented in Figure 6 and the different patterns observed in dD compared with d18O, I wonder if the authors could speak to the relevance of memory. What order were the standards analyzed in and were the vapor concentrations always stepped through from high

to low or low to high? Or did the authors rule out memory and were left with suggesting spectral fitting. If the pattern is spectral, I wonder if the 18O of laser 2 shows a different pattern compared with the 18O of laser 1. What about 17O? Does it show a unique pattern?

The reviewer is absolutely correct that memory needs to be taken into account when carrying out the humidity-isotope response curve. In fact, we believe this is an often overlooked source of error. It is therefore important to make sure that memory has been removed before starting the humidity-isotope-calibration. We believe that we did this with care. Laboratory tests have showns an advantage that once you have removed the memory effect, you can get a more stable humidity isotope-response curve when change the humidity from low to high. The effect of this, is only present when you have ensured that there is not influence of memory effect. We do note that the difference in humidity isotope response as function of water isotope value is a consistent result across our combined laboratory and field experience with multiple different instrument analyzer. If the reviewer has a suggestion for testing this further and understanding the course of this difference in humidity-isotope-response we will be very happy if the reviewer contacts us off-line for discussing tests to be done and instrument performance values to be extracted.

---

## Author Comment (AC4)

**Response to Reviewer 4 on review of "A versatile water vapor generation module for vapor isotope calibration and liquid isotope measurements"**

The reviewer has provided the following review. However, as our paper does not deal with any UAV systems or EC systems we believe that the review carried out by the reviewer was intended for a different manuscript. We are hence not able to respond to the reviewer.

**1) Elucidate that what are the advantages or improvements of the UAV-based EC system developed by authors over other existing UAV-based flux measurement systems.**

**2) Show that what are the differences or improvements in the calculation method of wind or turbulent flux for the current UAV-based EC system compared with manned airborne EC systems.**

**3) Measurement precision or reliability is an important metric for the successful application of the UAV EC methods, the current manuscript only gave the mathematical precision (or instrumental error) in measurement of wind and turbulent flux. I recommend that the authors could make a direct comparison between the measurement from UAV- and ground-based EC systems.**

---

## Author Comment (AC5)

**Response to Reviewer 5 on review of "A versatile water vapor generation module for vapor isotope calibration and liquid isotope measurements"**

We thank Reviewer #5 for valuable inputs, which have improved the readability of the manuscript. As suggested by the reviewer we have placed our results into a broader perspective by comparing our 17O-excess measurement performance with what we believe is the most recent work such as Davidge et al. 2022. We have responded with red font below.

The paper reports general improvements in the author's previous device. I found it challenging to understand the manuscript in detail simply because it is not written for broader readers. Too many self-citations (14/43=32%) imply a narrow target readership. It would be preferable to clarify and discuss the position of this study from a broad perspective, as related studies exist. For example, Graaf et al. (Isotope ratio infrared spectroscopy analysis of water samples without memory effects, Rapid Com. Mass, 2021) reported no memory effect for liquid measurement. Although these test results would be helpful for certain people, this manuscript should be thoroughly revised before acceptance.

We agree with the reviewer that the study of Graaf et al. is very interesting, especially the result focused on the reduction in memory effect. The setup of Graaf et al. follows the development of Gkinis et al. 2011, which uses a peristaltic pumpt to push water into a heated tee. We note that we already reference this development in our introduction. Graaf et al. then injects a sample using an autosampler into a heated chamber designed similar to the Picarro High Throughput vaporizer or the LGR vaporization system. However, the instrumental development of Graaf et al. suffers from the same design issue as the Picarro SDM or Picarro/LGR autosampler/vaporizer in that that it does not allow an unlimited injection of a sample as described in section 2.1 Graf et al. "*This volume serves to spread out the sample peak over a wider time interval, which facilitates precise data integration over the sample peak.*"
We believe that the continuous injection of a sample or a standard such as documented for 92 hours is a key development of our system. The long injection time allows us to quantify the Allan Deviation for integration for more than 24 hours which allow us to detect optimal injection times for obtaining low standard deviation levels. The analysis also allows us to compare the performance of the two systems.
For example: We report an Allan Deviation on 17O-excess on 10 minute averaging time to be around 10 per meg. This value should be compared to the 1-sigma precision on 17O-excess for a single injection presented by Graaf et al which is shown to be 60 per meg.

Graaf et al report a standard error of 10 per meg for 17O-excess:
*"In the system presented here, the statistical uncertainty on $\Delta'17O$ can be reduced to the 10 per meg level by determining standard errors of the mean of 20 single injections, which corresponds to 3.5 h analysis time per sample."*

Using similar measurement times, we show that the reproducibility for samples measured 4-5 times (over 3 hours each time) on separate days reveals a standard error of the mean always ≤ 4 per meg. (It is important to note that the precision reported in our manuscript is defined as the standard error of the mean multiplied by the Student's t-factor for a 95% confidence limit, as described in Barkan and Luz (2005).

However, we do note that our study is not the only recent study presenting similar low errors on 17O-excess. We have therefore added the following sentences to place our result in perspective: *Such reproducibility is comparable to precision achieved with IRMS (e.g. Barkan and Luz, 2005; Steig et al., 2014) and to the total error obtained within the latest development in continuous flow analysis of ice cores (Davidge et al., 2022)*

 (1) There are many typos. I cannot point out all. Following are the typos found only on page 1.

We agree that this should have been fix by us and we have therefore gone through the manuscript carefully to remove any issues. Hopefully, it is much better now.

L17: show > shows
Fixed

L22: Remove "as" at the end of the sentence.
Fixed

L23: have > has
Fixed

L23: document > documenting (or documented?)
Fixed

L30: falls > fall
Fixed

(2) An example of the limited readability only on Figure 1.
New schematic done following reviewers comments

L.132: "...module is an improved and revised version of an original prototype in 2014…"

> Please briefly explain the previous system. A few sentences will be fine.
Fixed

L. 134: "A four-ovens version was used in this study."

> What is a four-oven version? Why does Figure 1 illustrate only one oven?

We agree that Figure 1 perhaps was not so clear on this point, as Figure 1 illustrated "an oven unit", which was then further described in Figure 2. We have therefore updated Figure 1 to show the individual ovens as well as provided a more detailed drawing of the oven themselves in Figure 2.

In Fig.1, caption: "… in open-split mode…"

>Please explain the open-split mode.]

We have updated the text so that it now reads "...of the vapor generation module in open-split mode to allow the part of the vapor stream that is not going into the Picarro instrument to escape"

Fig. 1: There are no explanations about abbreviations and lines. Please define or explain "P1", "T1", "Tn", "Vacuum line", "red dotted line", "black dotted line", and "black solid line ".

We have prepared a revised version of Fig. 1. All the lines are now color-coded and a legend is included in the figure. Reference to the vials ($T1…Tn$) has been removed. The schematics now include reference to the product number of the valves.

Specifically we have added the following information to the supplementary material and referenced it in the caption of Figure 1:

$V1\_x$ are manifold-mounted three way solenoid valves (SMC, VO307-6DZ1-Q, VV307-01-043-01N-F)

$V2\_x$, $V3\_x$, $V4\_x$ are manifold-mounted solenoid valves (SMC, VDW23-6W-1-G-Q, VV2DW2-G0401N-F-Q)

Humidity sensor is obtained by combining temperature sensor (Analog Devices, AD22100STZ) and relative humidity (Honeywell, HIH-4000-004).

MFC1 and MFC2 are mass flow controllers for air (Aalborg, model GFC17A)

Dual-valve electronic pressure controller is ALICAT model PCD-5PSIG

Vacuum pump is KNF membrane pump model NMP830KPDC

PID control is achieved by LabVIEW software.

Please add several photos of this system in supplementary information.

We have added figures to the supplementary material under Supplementary Images.

---

## Author Comment (AC6)

**Response to Reviewer 6 on review of "A versatile water vapor generation module for vapor isotope calibration and liquid isotope measurements"**

We are grateful for the comments and suggestions provided by Reviewer #6, which are truly valuable and insightful. The comments have made us reflect on how to improve future versions of both the vapor generator and the control software. We have below responded to the reviewer using red font. All 17Oexcess values have now been calibrated as suggested by the reviewer.

General Comments

The manuscript details a field-deployable custom vapor generation system that will facilitate field measurements of water isotopes; the analysis demonstrates the analytical advantages of the method relative to vaporization units that rely on discrete injection. The manuscript will benefit readers of AMT who utilize water isotope analysis systems in the field and laboratory. The authors claim that the custom unit will benefit field measurement campaigns because the vaporization unit is portable, easy to service, and can measure reference waters more quickly than commercially available units and with less isotope memory. The authors tested the system and show that it meets analytical precision targets. They also conduct a stability test of the vapor generation module by measuring the vapor stream with two Picarro instruments simultaneously to show that the noise from the vaporizer system is small relative the noise of the instruments. More information about operational protocols and maintenance of the vaporizer unit will help convince a reader of its field-worthiness.

Thank you for this advice. As we described in the manuscript we had partial success to unclog the capillaries using an ultrasonic bath. We have also tried to push citric acid through the capillaries, but it is not always working. We have since the submission of the manuscript found that using in-house prepared stands either from miliQ water from Bermuda or a mixture with Greenland snow allowed us to run the system in the laboratory every day for a 6 months period. We have updated the text about the length of operation in the manuscript such that it now reads: "We have used the calibration system in the laboratory and during field campaigns for about 2 years now and found that a stable performance of the vapor generation module is dependent on using clean standards. When using in-house generated standards consisting of a mixture of melted Greenland snow and milli-Q water from Bermuda we have successfully operated the vapor generation module daily (roughly 1-3 hours every day) in the laboratory for more than 6 months without changing the capillaries."

The apparent relationship between cavity temperature fluctuations and d18O on timescales of ~tens of minutes is an important finding of this analysis and should be highlighted; as the authors suggest, if the instrument temperature could be stabilized, this would improve precision for d18O, but this also has likely significant implications for the deuterium- and 17O- excess measurements as well.

Thank you. We agree that this is an important finding and have tried to stress this in the manuscript text and in the abstract:
We write in section 4.1
"We also further notice that this increase in Allan Deviation occurs at similar integration times as used for liquid measurements, which makes it even more important to improve on."

And in the Abstract:
"Using the vapor generation module, we document that an enhancement in the Allan Deviation above the white noise level for integration times between 10 minutes and 1 hour is caused by cyclic variations in the cavity temperature, which if improved upon could result in an improvement in liquid sample measurement precision of up to a factor 2"

**Reorganization and reprioritization of some discussion points would improve this article.**
As it is written, the abstract does not summarize or highlight the most important outcomes of the study. Instead of focusing the abstract on theoretical future applications (i.e. instead of saying what it "could in principle" do), please revise the abstract to **document the novel advantages of the vapor generator and highlight the new results that are achieved by the tests described in the manuscript**.
We have since the submission of the manuscript been able to operate the instrument for more than 6 months without the need to do any maintenance. It has therefore been possible for us to update the abstract to now read:
"The vapor generation module as a calibration system has been documented to generate a constant water vapor stream for more than 90 hours showing the feasibility of being used to integrate measurements over much longer periods than achievable with syringe-based injections as well as allowing the analysis of instrument performance and noise. Using clean in-house standards, we have achieved to operate the vapor generation module daily for 1-3 hours for more than 6 months without the need for maintenance, illustrating the potential as a field-deployed autonomous vapor isotope calibration unit. "

We have also removed the sentence including the "could in principle" when dealing with performance, but not when it comes to number of standards/samples to be connected as it does not make sense to give an actual number as the restriction is physical space.

The vaporizer module is of course important for its potential to field-calibrate, but it is also useful for the analysis of instrumental noise, examining isotope-humidity dependence, and other performance metrics that are shown in the paper and deserve emphasis. For example, the abstract could perhaps highlight the vaporizer's ability to generate continuous water vapor for a wide range of humidity levels (which is an advantage for vapor measurements over CFA vaporizer

configurations like Gkinis, Jones, or Davidge that target a single water vapor concentration) with the good stability at each level but increasing memory issues at low humidity.

Thank you for this suggestion.. We have now updated the abstract to read:

"The vapor generation module can generate a stream of constant vapor at a wide variety of humidity levels spanning 300 ppmv to 30 000 ppmv and is fully scalable allowing in principle an unlimited number of standards or samples to be connected. This versatility opens up the possibility for calibrating with multiple standards during field deployment including examining instrument isotope-humidity dependence."

We do not find evidence that the vapor generation module shows an increasing memory effect at low humidity. Perhaps the reviewer is referring to decreased relative humidity stability at low humidity. This could perhaps be explained by the higher dilution flow rate, which could increase the pressure difference between the capillary and the open split. Should variations in this pressure gradient exists due to MFC instabilities there would be variations in the delivery of the liquid into the oven through the capillary leading to humidity variations.

We have further more include a figure (Figur 5 in revised manuscript):

[Figure]

Illustrating the stability of the humidity levels during a humidity-isotope calibration.

The comparisons with discrete injections (such as in Figure 4) are important documentation of the advantages of continuous over discrete vaporization, but do not uniquely reflect the contributions of this vaporization unit over other continuous vaporizers designed for CFA – the abstract (and manuscript) could better contextualize the new contributions made by the authors with this custom vaporizer through direct comparisons to other published calibration methods and continuous vaporizer units.

We have improved the abstract with the following sentence:

The vapor generation module can generate a stream of constant vapor at a wide variety of humidity levels spanning 300 ppmv to 30 000 ppmv

This versatility opens up the possibility for calibrating with multiple standards during field deployment including examining instrument isotope-humidity dependence. Utilizing the ability to generate an uninterrupted constant stream of vapor we document an Allan Deviation for $^{17}$O-excess ($\Delta^{17}$O) of less than 2 per meg for an approximate 3-hour averaging time

Using the vapor generation module, we document that an enhancement in the Allan Deviation above the white noise level for integration times between 10 minutes and 1 hour is caused by cyclic variations in the cavity temperature, which if improved upon could result in an improvement in liquid sample measurement precision of up to a factor 2.

The vapor generation module as a calibration system has been documented to generate a constant water vapor stream for more than 90 hours showing the feasibility of being used to integrate measurements over much longer periods than achievable with syringe-based injections as well as allowing the analysis of instrument performance and noise. Using clean in-house standards, we have achieved to operate the vapor generation module daily for 1-3 hours for more than 6 months without the need for maintenance, illustrating the potential as a field-deployed autonomous vapor isotope calibration unit.   When operating the vapor generation module for laboratory-based liquid water isotope measurements we document a more than 2 times lower memory effect compared to a standard autosampler system.

Finally, the authors aim to demonstrate that the custom vaporizer has sufficient signal stability to measure 17O-excess, but **the authors need to be more cautious about the treatment of raw data to assess performance for 17O-excess**. It is misleading to assign units of per meg to the raw data since the raw signal variability is not equivalent to the calibrated signal variability. This treatment potentially affects data shown in figures 3, 5 and certainly in figures 8(i,j) and S2.

We agree with the reviewer that raw signal variability is not equivalent to calibrated signal variability. For this reason, we ran all the analysis and prepared new plots using calibrated data. The calibration lines were defined during the SP BER step change in the 90 hours long run. The system was primed with standard water vapor for 2 hours and the measured $\delta^{17}$O $\delta^{18}$O $\delta$D were defined by averaging 1 hour of data after the priming. The new Figures 3 and 5 have changed in the manuscript, accordingly. It should be noted that changes in the plots are extremely small because calibration affects only to a minor extent the signal variance (i.e. the slope of the calibration line is ~1 ‰/‰).
Following suggestions by reviewer #6, we also have edited Figure 8, which now reports the variability of the raw signal for each pulse around the mean calculated from all the pulses.

Previous work has established accuracy (not precision) as the dominant error in CRDS 17O-excess data, so while the precision seems great for these data, more work is needed to show that system operating conditions like automated, variable flow rates or secondary air dilution do not create calibration bias for 17O-excess. **Though small, the systematic offset in the calibrated 17O-excess data further suggest that there likely is a bias in the calibration, which should either be examined further to better characterize the limitations of this method or should be qualified appropriately in the text when claims are made about the quality of 17O-excess data.**

We agree with reviewer #6 that accuracy in 17O-excess is more problematic to handle rather than precision, since our manuscript and others highlight very high stability for 2140 models. Specific guidelines and *certified* 17O excess values to perform laboratory calibration should be provided by IAEA to this end. With the tools in our hand, we are only able to account for a potential error introduced by the weighting procedure for the preparation of the mixtures. This is described in the text as the maximum span obtained by mixing higher amount of the first standard (+0.01g) and smaller amount of the second standard (-0.01g), and vice versa. This conservative estimate of the sample uncertainty yields a 13 per meg variation of the final Δ17O value, which is actually larger than the inter-run precision we obtain.

We have further added the following line: "...but within the expected uncertainty due to the weighting uncertainty, which could produce an off set of up to 13 per meg."

Specific Comments

Some sections of the text need to be revised for clarity and completeness. In some sections, the claims that are made are not directly supported by the evidence provided in the tables/figures so it can be difficult to evaluate some statements. The whole manuscript would benefit from a careful reread and review by the authors.
We acknowledge that too many grammatical errors occurred.

The abstract claims that the vapor stream is constant for 90+ hours and that it therefore could be used in the field for more than three months, but it is unclear what "constant" means in this context, especially since the water vapor data that is shown in figure 3 exhibits some notable variations. It is also unclear (in the abstract) how operating for 90 hours in the lab translates to three months in the field – this is explained at the very end of the manuscript (in that it will theoretically be measuring standards for 1h/day) but is confusing in the abstract since it is not explained. **Further, the 90hr to 3 month relationship is speculative at best, since it has not been demonstrated that the unit will, for example, not clog with precipitates or encounter**

**other operational setbacks over the three month window.** More information about typical or intended operations of this system will help the reader better understand its advantages and limitations.

As described above we have updated the claim based on the operations since submission of manuscript:
"We have used the calibration system in the laboratory and during field campaigns for about 2 years now and found that a stable performance of the vapor generation module is dependent on using clean standards. When using in-house generated standards consisting of a mixture of melted Greenland snow and milli-Q water from Bermuda we have successfully operated the vapor generation module daily (roughly 1-3 hours every day) in the laboratory for more than 6 months without changing the capillaries."
We further have added the sentence at the end of section 4.3 illustrating the importance of using clean standards:
While our experience using our in-house clean standards shows stable performance by the vapor generation module, we have also used standards, which were provided to us by other labs. In those cases, we observed that the capillary would get clogged more frequently and had to be replaced every 2-4 weeks. To extend the lifetime of the capillaries it is possible to use a larger ID, but this can potentially results in less stable humidity values.

When using the vapor generation module for unknown liquid sample measurements it is hence likely that one need to include a change in capillary as part of operational procedure. While not discussed in the manuscript there is a price advantage of the capillaries (~10  EUR each) compared to syringes used in an autosampler (~100 EUR each)

There is a lot of confusing or vague language throughout the document and also many acronyms that have not been defined – please try to define acronyms before using them in the text or figure captions and make sure details of system components are explained the first time they are mentioned.
We have done this now for all acronyms.

Ln 53-55 suggests that excess values require a "relatively high output of individual number of samples measured" – can you clarify what you mean here?
What we meant was that the use of d-excess and 17O-excess still require a relative large number of samples to be measured in order to make robust conclusions, but we agree this is not clear in text and we have therefore removed that part of the sentence.

Ln 72-75: define "low uncertainty" and "large quantities".

Corrected:
(providing calibration pulses with an uncertainty of +/- 0.1/1.0 ‰ for d18O/dD)

And
(3-5 litre depending on deployment period)
Ln 91: Davidge et al 2022 utilizes a unique vapor generation system so "this system" should instead say something like "a similar system".
Indeed - this is corrected now

ln 151: instead of noting the differences between this vaporizer and an earlier version of this vaporizer, this might be a good place to describe the differences and advantages of this system over other types of vaporization systems (e.g. it adopts the multi-channel selector valve of Jones et al. 2017, similar continuous vaporization setup to Gkinis but with vacuum pump, PID control for humidity, additional mixing tee, etc.).
Reviewer #5 have asked us to further expand on the differences between this vapor generation module and earlier versions hence we would like to keep this list drawing attention of the reviewer on the detailed improvements.
We believe that we describe in the section under the bullet point list the reasons for each addition as well as also references Jones et al. 2017 for the inspiration to connect a selector valve.
It is of course also clear, as we hope is illustrated by our referencing that we are building on the last decade of CFA and vapor calibration developments.

Many of the details noted on page 6 would be better left to the later sections of the paper since a general reader might not understand the specifics about salt deposits, number of ovens, valve circuitry, etc. at this point in the paper.
We would prefer to keep this detailed section under Methods and leave the Discussion section to performance.

Ln 190: if the pressure regulator resolution is 0.01psi, why is the regulated range of pressures that is listed so large (0.5-3.5psi)? How is this system typically operated for each humidity level, and how much does the pressure fluctuate for each humidity level during a typical measurement? More operational details are needed to help a reader duplicate this work.
0.01 is the resolution of the electronic pressure controller but to observe a significant variation of humidity (>100 ppmv) the headspace pressure needs to be changed in the order of 0.1 - 0.2 psi. Typical pressure applied using a new capillary for generating humidity levels around 5-10kppmv is around 1 psi. However, as the capillary becomes clogged the pressure can be increased to 2 psi. An experienced operator can also start out by a relative high pressure of 1.5-2 psi to initiate the flow of water through the tube and then reduce the pressure at the first sign of water being delivered to the oven. However, one need to be careful about pushing too much water into the oven and hence flooding it.

**We haven't log the pressure fluctuation in the headspace, maybe we can show a step change in headspace pressure and step change pressure in humidity in this document?**

All tables and figures should have sufficiently detailed captions so that the reader can easily understand what data is shown – please reread all figure and table captions and try to add more information.
We have tried to fix this

Ln ~265: perhaps it would be advantageous to highlight the regions of the plot that the author is describing in the text when talking about the slope of the allan deviation with time.
Due to the fact that white noise seems to be dominant for different periods for d18O compared to dD we think it will be difficult to illustrate a specific region, when it comes to white noise. We have, however indicated the expected slope for a decrease in Allan deviation when white noise being dominant.

Ln 271-272: figure 3 is important both because it documents the ability of this vaporizer unit to generate consistent isotope data, but also because the authors have highlighted the difference in the allan variance analysis when truncating the data to account for memory effects within the system. The vaporizer unit exhibits excellent performance and this should be highlighted, but it is incorrect to say that the performance is better than that of other systems, since other systems show similar precision at these averaging times (e.g. Gkinis et al. 2010 for dD and d18O, Steig et al. 2021, or Davidge et al. 2022 for 17O-excess). It is therefore also incorrect to suggest that no previous work has managed system memory effectively.

We agree with the reviewer that to a large extend the performance for the vapor generation unit presented here show similar performace as Gkinis et al. 2010, Steig et al. 2021, and Davidge et al. 2022. However, there are some small but important differences.

For dD we obtain a minimum in Allan Deviation at 1.5e4 s with an Allan Deviation of 1e-2 permil.
In Gkinis et al. 2010 the minimum is achieved at 3e3 s with an Allan Deviation of 3e-2 permil. After integration time 3e3s the Allan Deviation deviates from the -0.5 sigma_allan/Tau, white noise line, indicating either drift of the instrument of not completely removed memory effect. Comparing to our estimated Allan Deviation at 3e3s we obtain 2e-2 permil, which is interesting since it indicates that the difference between a 2140 and 1100 series picarro is a 30% reduction in noise for dD ( For d18O the difference is 8e-3 vs 5e-3 at 7e3s integration time). As the d18O Allan deviation curve seems to continue decreasing after the dD curve has become constant seems to indicate an issue with memory effect.

For Steig et al. 2021 the Allan deviation for dD seems to deviate from the white noise line already around 1e2 seconds, while d18O continues to decrease down to 6e3 seconds. However, the "bump" hypothesized being driven by cavity temperature is occurring between 1e2 and 8e2, which also has an effect. However, it seems that memory effect is influencing the results.

For Davidge et al. 2022 no Allan deviation for d18O/dD is shown, but instead shows an allan deviation plot for 17O-excess. The Allan deviation plot hower does not extend past 6e3 s integration time. In our manuscript we present an Allan Devation plot extending to 8e4 second integration time - documenting a minimum at 2e4 second integration time at 1.3 per meg. Davidge et al. 2022 obtains a minimum of 3 per meg 3e3 second integration. This is comparable to our results.

We do believe based on the above argument that our system is presenting an improvement.
We have however removed the "suboptimal"-word here.
We have also added the following sentence: "Albeit for $10^3$ seconds, we show similar $\Delta^{17}O$ $\boldsymbol{\sigma}_{Allan}$ as Davidge et al. (2022)"

The authors might consider showing the data from Table 2 as a figure, or perhaps showing the full sequence of measurements made over time to help the reader understand the tests that were conducted.
We agree and have added the following figure.

[Figure]

The 12500 ppmv threshold finding is very interesting, especially since the allan variance test shows the worst performance in this region. Have you tested whether this is a physical effect of the capillary diameter or some other design choice? Were all data generated for this study using the 127um capillary?

We have unfortunately not carried out a complete evaluation of all capillary diameters.

Ln 397: this replication looks promising, but more information about the calibration is needed here. This section should also include that this performance is comparable to other continuous vaporizers that have been developed for 17O-excess measurements (i.e. Steig 2021, Davidge 2022).

Following our answer to reviewer #3, a typical measurement session (run) is composed of analysis of two standards followed by analysis of two samples, as shown below. Independent calibration means that each run is calibrated using the two standards of that specific run. Average calibration means that the results of the analysis of the standards have been averaged and an average calibration line was built to calibrate the analysis of the samples.

[Figure]

This procedure is now described in the text as follows:

*Since the standards were measured from 2 to 4 times for each measurement session, and the solutions were measured on different days, we applied both an average and independent calibration. The average calibration factors were estimated by averaging the raw observations of the standards injected for all the run. Instead, the independent calibration factors were calculated for each couple of reference standards and then applied only to the following couple of solutions (2 standards followed by 2 samples). The results of the experiment are reported in Table 3.*

Also the following text has been edited:
*Such reproducibility is comparable to precision achieved with IRMS (e.g. Barkan and Luz, 2005; Steig et al., 2014) and to the total error obtained within the latest development in continuous flow analysis of ice cores (Davidge et al., 2022) but better than analysis performed with optimal settings of vaporizer and autosampler from Picarro (which is 8 per meg following Schauer et al. 2016).*

Ln 401: The offset is likely due to calibration, so additional details about how these data were calibrated would be useful. It seems unlikely that the age of the reference water has modified the 17Oexcess values if they have been in sealed containers and cold storage, but it is possible that the systematic bias is due to an error in the calibration standard assignments or that it is generated by the vaporizer unit itself, which must be carefully examined. Which standards were used for calibration? How frequently were they measured? How stable were the raw values in d18O/d17O between reference water measurements? Without this information it is impossible to speculate what might be the cause of these offsets.

We are not completely sure about the origin of the offset. We suggested an error in the weighting procedure as a potential source of uncertainty but also error in the calibration standard assignments is a potential uncertainty, as the referee suggests. In principle we should send our standards to another lab and perform a lab-intercomparison, which is however at the moment is out of the scope of this manuscript (here we focus on the precision and repeatability of the measurement).

We believe that a bias generated by the calibration system itself is less likely, since flash evaporator is a well-proven technique for water stable isotope analysis. Moreover, we performed internal laboratory calibration using VSMOW-SLAP2 using both the Picarro's liquid injection mode and the calibration system and we have obtained identical d18O, d17O, dD values for the two methods (the data is not reported in the study).

Answers to specific questions in the comment.

**Which standards were used for calibration?** The standard used were always SW and WW, the same standards used to produce the mixtures.

**How frequently were they measured?** The standards were analyzed every two samples (Please see figure attached for Reviewer #3. That means the standards were injected between 2 and 4 times for each run.

**How stable were the raw values in d18O/d17O between reference water measurements?** 1 standard deviations (SD) and the standard errors of the mean (SEM) of raw reference water measurements across all the runs are reported in the following table. Values in permil (‰).

|  | $\delta^{17}O$ (SD) | $\delta^{17}O$ (SEM) |  | $\delta^{18}O$ (SD) | $\delta^{18}O$ (SEM) |
|---|---|---|---|---|---|
| **SW** | 0.026 | 0.009 |  | 0.037 | 0.013 |
| **WW** | 0.017 | 0.006 |  | 0.032 | 0.011 |

Throughout the paper and abstract there are speculations about what could be done in principle, but it is important to properly document what can be done in practice, especially since these systems do require maintenance and cannot run indefinitely or measure standards frequently enough for perfect calibrations. How often is it necessary to clean the capillary? What maintenance was required for this system during the study period and with what frequency? How can one best operate a system like this within those maintenance limitations to maximize the quantity and the quality of the data? Details about operational controls, conditions, and maintenance would help the reader better understand the performance of this vaporizer system.

We have improved the text based on our experiences over the last 6 months running the system daily in the laboratory. We have also updated the manuscript to illustrate the need for use of clean standards. Specifically:

We have used the calibration system in the laboratory and during field campaigns for about 2 years now and found that a stable performance of the vapor generation module is dependent on using clean standards. When using in-house generated standards consisting of a mixture of melted Greenland snow and milli-Q water from Bermuda we have successfully operated the vapor generation module daily (roughly 1-3 hours every day) in the laboratory for more than 6 months without changing the capillaries.

While our experience using our in-house clean standards shows stable performance by the vapor generation module, we have also used standards, which were provided to us by other labs. In those cases, we observed that the capillary would get clogged more frequently and had to be replaced every 2-4 weeks. To extend the lifetime of the capillaries it is possible to use a larger ID, but this can potentially result in less stable humidity values.

Ln 465 should also acknowledge the increase in the delta values.
Yes - we have updated the text

Ln 469 seems like a major limitation of this method – perhaps the authors should repeat this test with larger vials to eliminate this enrichment issue.
We agree and have learned from our mistake. By including this illustration in the manuscript we hope that anyone doing similar test will not make similar mistake.

Ln 480: please define "relatively clean"
Yes - have updated the discussion to say that using milliQ water as standards did not provide any clogging within 6 months.

Ln 483: is 1h per day sufficient for the calibration of all water isotopes? Why has this duration been chosen?
We have removed the reference to 1 hour per day calibration in the manuscript. However, from our experience to obtain high quality d18O and dD water vapor isotope measurements a rule of thumb is to calibrate for 1 hour per day.

The discussion in ln ~510 and data shown in Figure SM2 suggest that the performance of this vaporizer is changing over these 48h of analysis due to the automation of air/water ratios and necessary reduction in flow rate to accommodate the formation of salts in the capillary. Have you tested this system with milli-Q or other treated water to minimize this effect, and have you seen any improvement in this performance? A change of ~3 per mil dD over this short timescale should probably be investigated further. The way this is accounted for in CFA systems is by keeping the liquid/air injection rates as constant as possible, because otherwise it is impossible to

know what the effect of the memory is at any point during the analysis when the flow rates are changing during analysis. What range of flow rates does this study utilize and can you attribute any of the changes in system performance to these variables?

We were in fact debating whether to include the figure SM2 and the discussion in the manuscript or not as we do not think that such a change in measured standard values provide an appropriate image of the performance of the vapor generation module that we wish to convey. However, we chose to include this illustration as it provide a good example of how memory effect can influence the measurements of the standards or samples. In other words: by illustration we hope people will be aware of the pit-falls.
We appreciate the suggestion on keeping the liquid/air rates constant and we are planning on logging the flow rate of the air in any follow up versions of the vapor generation module.
Not directly relevant to the discussion here, but we have been analyzing the long-term memory effect of the Picarro High Precision Vaporizer by logging the memory correction values provided by the Van Geldern method. We discovered a slow by increasing memory effect of the vaporizer, which potentially can have influence on long term measurement performance. Hence, the issue with drift in memory effect is not only an issue for our system.

Ln 521: define "relatively high measurement uncertainty" and other vague quantities throughout the paper.

The manuscript has been checked and edited accordingly using numerical quantities. Specifically for Ln 521, the text now is:

*Due to the relatively high measurement uncertainty (e.g. $\sigma_{Allan}$ = 9 per meg @ 15 minutes) and relatively small changes in $\Delta^{17}O$ observed in natural waters (~90 per meg), it is not the memory effect, which is the limiting factor influencing the $\Delta^{17}O$ measurements.*

Ln 523: change "error, which is" to "error that is" for clarity/correctness.
Corrected

Ln 524: without additional analysis of calibrated data it is a stretch to say that the module is "optimal for 17O-excess" but it is certainly promising to see such nice signal stability in the 17O-excess record. Previous work has established calibration as the major limitation on laser spectroscopy measurements for 17O-excess, so without additional analysis of calibrated data it is hard to accept these claims.
OK - we have changed the wording to "is highly capable of 17O-excess measurements"

Ln 525: similarly, while the unit can operate over long periods, unless it is possible to measure sufficient durations of the calibration standards for 17O-excess in between vapor samples, the resulting data could have large errors – this issue is discussed in both Steig et al. 2021 and Davidge et al. 2022. How long can the vaporizer operate before the capillary clogs or the flow rates change? This data would be important for understanding limitations around 17O-excess calibration.

OK we have updated the text to be more modest:

"Hence, the vapor generation module is highly capable of $\Delta^{17}O$ measurements as it can provide integration times over multiple hours of both standards and samples allowing optimal treatment of memory effects and measurement noise to be reduced to a minimum.
"

Though I look forward to following your updates in the future, Section 4.5 is not analysis of the data that is presented in this paper and could be better suited for a proposal.

OK - we have shorten the text to only discuss planned improvements for sample measurements.

Table and Figure Comments

Figure 1 – this will be difficult for many readers to follow. I suggest additional labels and defining acronyms and process control symbols.

DONE

Figure 2 – recommend additional labels to help the reader understand this figure

DONE

Figure 3 – clear figure. Can you comment on the variability in the water vapor concentration? Especially since earlier studies have linked water vaporization inconsistencies to isotope fractionation it seems important to better characterize these vapor fluctuations.

Figure 4 – great visualization of this relationship that shows a major advantage of continuous injection modes for laser spectroscopy. The legend is perhaps a little confusing because it is unclear what the legend means without also reading the caption.

DONE

Figure 5 – Why not show all water isotopes here? Also please consider revising the label on the y axis if the data used for this analysis has not been calibrated.

Additional plots in SM

Figure 6 – great figure with important implications for low-humidity measurements. Maybe instead of defining d18O_diff in the caption you could just label it d18O_3500ppmv – d18O or similar?
DONE

Figure 7 – if possible, moving the legend away from the data would make it easier to read panel A. This is a very compelling and disturbing result!
DONE

Figure 8 – the authors should either calibrate the 17O-excess data or find another way to describe the spread of the raw spectroscopy response – showing values of 300 per meg is very misleading! Large variability in the 17O-excess raw data suggests that the errors in d17O and d18O are not perfectly correlated, which is likely to cause accuracy issues in the calibrated 17O-excess values. Because the data is not calibrated it is difficult to evaluate this data – please provide a record of calibrated 17O-excess over time in the revision of this manuscript.
DONE, normalized

Figure 9 – this figure would be more useful if calibrated 17O-excess values were shown.
Here is the 17O-excess step change (calibrated)..

[Figure]

.

Table 1 – define BER and SP. What is the uncertainty of the values of the excess measurements and the SP measurements? Where were the data measured? Please specify whether these values are measured relative the VSMOW-SLAP scale. It would be helpful to combine Tables 1 and SM1 and show all waters here since the reference waters from the supplement are referred to in the text.

The standards used in this study have been provided by different stable isotopes laboratories (Laboratoire des Sciences du Climat et de l'Environnement, Centre for Ice and Climate at the Niels Bohr Institute, and the Stable Isotope Laboratory at the Institute of Arctic and Alpine Research, University of Colorado). Not all the reference values were provided with an uncertainty. We know that providing a reference value without any uncertainty is not best practice, but internal standards are subject to uncertainty and quality checks that are laboratoryspecific. We treated all the standards as true values assuming no knowledge about their uncertainty. Hence, we removed the uncertainties from Table 1 for consistency.

Table 2 – as noted above, a plot of this data could be helpful for the reader to understand the different tests that were conducted. Is the standard deviation calculated for the raw data over the full duration of each test?

Also following the comment of the other reviewers, Table 2 has been converted into a figure. The table is now reported in the supplementary material.

Table 3 – please include details about how these measurements were calibrated.

Following our answer to reviewer #3: a typical measurement session (run) is composed of analysis of two standards followed by analysis of two samples, as shown below. Independent calibration means that each run is calibrated using the two standards of that specific run. Average calibration means that the results of the analysis of the standards have been averaged and an average calibration line was built to calibrate the analysis of the samples.

[Figure]

This procedure is described in the text as follows:

*Since the standards were from 2 to 4 times for each measurement session, and the solutions were measured on different days, we applied both an average and independent calibration. The average calibration factors were estimated by averaging the raw observations of the standards injected for all the run. Instead, the independent calibration factors were calculated for each couple of reference standards and then applied only to the following couple of solutions (2 standards followed by 2 samples). The results of the experiment are reported in Table 3.*

Table S2 – Please check for rounding error in the 17O-excess reference water assignments; from the isotope values and mixing ratios given I calculate 20, 11, and 17 per meg (not 21, 12, and 18), though this does not substantially change the result or interpretation.

There was a typo in the digits of d17O. However, the new values calculated are: 21,12,19 (only this one is different from the previous calculation).

This is the formula I have used:

(ln(**d17O**/1000+1)-0.528*LN(**d18O**/1000+1))*1000000
Also Table 3 in the manuscript must be changed.

Figure S1 -  the memory effect for dD appears to be different during these two analysis windows; it could be interesting to examine the operating conditions during these runs and consider whether flow rates or other changed conditions could cause this difference.
Unfortunately we did not log the dry air flow rate, which will be key to be done in our future software updates.

---

## Referee Report (RR1)

**Comments on memory and allan variance**

Especially considering that this paper demonstrates that the high-frequency variability is owed to the stability of each instrument and not the vaporization module itself, these plots all fall well within the range of values published by Steig 2021, whose data are overlain on Figs 3 and 8 below and deviate notably only at high-frequencies. The Steig 2021 data are from much shorter runs of calibration standards during the South Pole ice core measurements, and they achieve similar results for all isotopes, especially when compared with the older instrument in figure 8 or the untruncated data in figure 3. The Steig 2021 dD values are similar to the performance shown in figure 8 and far exceed the un-trimmed data, even without the extensive 16-hour trimming. This actually suggests that the memory internal to this new vaporization system is more significant than the memory of the system used in that paper. Can you speculate about why this very long data trimming is necessary or comment on your data processing workflow for routine measurements given this limitation? Please ensure that your language around stability and memory is better placed within the context of this type of similar vaporization system.

[Figure]

[Figure]

**Technical corrections and minor comments:**

ln 12: I suggest rephrasing this to discuss the measurement of many samples instead of their "connection"

ln 17: "measuring unknown samples shows" is specific to 17O, right?

ln 18: the standard error is not provided -- perhaps this is a typo?

ln 19: "enhancement" is not quite right here. Maybe rephrase to talk about the increase in deviation or noise level instead?

ln 21: typo, should say "factor of 2"

ln 27: "achieved to operate" could just say "operated"

ln 41: typo after citation, should say "is classically"

ln 46-47: please clean up citation formatting and italics. why not use the typical notation for the deuterium excess?

ln 70-73: language is very confusing and inefficient -- please rephrase the first few sentences. A suggestion for ln72 is to say "...deployment is the bubbler system, which has been used continuously..."

ln 74: "and that there is minimal" could just say "and the minimal"

ln 78: maybe "is not feasible for many campaigns" would be clearer

ln 94: double-check the 17O-excess notation here and throughout for consistency

ln 97-99: please rewrite this sentence as it is very confusing to follow

ln 109: maybe "accuracy, we have further developed the patent application which was published in Steen-Larsen (2016)."

ln 115: could say "sufficiently high accuracy for D17O"

ln 115-130: it might be more intuitive to understand the purpose of the two case studies if this whole section is rewritten as a brief paragraph about objectives and how you tested them

ln 135 is a good example of how to refer to the patent for the first time -- consider revising ln 109 to be similar

ln 170: typo, should refer to 4.3. please check section references throughout.

ln 170 and section 4.3 talk about decline in humidity values over long timescales, but figure 5 shows that it sometimes increases. we find that depending on the air flow and water flow, the precipitates either clog the tee itself or the capillary, and depending on which flow is decreasing (air or water) the instability can cause an increase or a decrease in humidity. is this similar? it seems consistent with the values shown in figure 5.

ln 194: typo, "between 0.5 to 3.5 should" say "between 0.5 and 3.5"

ln 195: why not just say "with a 1.59mm PEEK union"?

ln 220: "frequent" should be "frequently"

ln 222: SW and WW haven't been defined yet -- please make sure these and other acronyms are defined before use

ln 265: please see comments above -- the performance shown here is excellent, but it is not significantly different from Steig et al 2021, which does not trim data as extensively as is shown here or have nearly as much time for each run. this (and the comparison with fig 8) suggests an increase in memory for this system, which should be discussed in more detail, if only to demonstrate that it can be managed for the applications promised.

section 3.2 compares the memory for this new system to the memory of the autosampler, which demonstrates that this method could be useful for lab measurements. But how is memory handled for field-calibration? What is a typical field calibration workflow?

ln 318: should specify that the short-term trend is for the water vapor concentration

ln 337: it seems like the variability in dD values at low-humidity ranges in Fig. 7 could also be influenced by the relatively longer memory of dD in the system – has this been investigated? Even if the liquid flow rate is the same at all levels because of the secondary mixing at TEE2, the retention time of water within the optical cavity itself should also worsen with decreased humidity which seems like it could contribute to this effect?

ln 364: many readers will have difficulty understanding the statement of D17O-->d17O conversion

ln 377: consider "values" or something more specific than "one" for clarity

ln 379: it is unclear what you mean by "error in the scale"

ln 384+: this section is describing the experimental setup to examine the high-frequency noise – consider restructuring to include relevant details in the methods section of the paper instead

ln 392: typo, "exceeding" should be "excess"

ln 410-11: typo? – this sentence doesn't make sense to me

ln 480-5: this does seem like a general problem for automated adjustments, though certainly not beyond characterization. Have you attempted to characterize the range of this effect for your system?

Ln 484/S3: which standards are used for calibration in each of the tests? It is not clear if it is the same every time and which of the standards listed in the supplement are used.

Ln 491: what is "relatively longer"?

Ln 494: is this memory limitation practical? Is it possible to shorten the tubing, or remove other dead volume from the system?

ln 520: relative discrete autosampler injections

ln 536: does not say what the standard error is (and looks copied from earlier section of the paper)

ln 537-9: citations would be helpful here and elsewhere – this should also be discussed earlier in the paper

table 1: how is 17Oexcess determined for BER? d17O does not have enough significant digits to determine these values.

table 2: I'm not sure there's an advantage to showing both sets of calibrated values -- perhaps just choose one calibration method, explain it in the text, and report the values in the table?

---

## Author Response (AR2)

Response to review by Reviewer #6

We thank Reviewer #6 for the detailed review. We have addressed the comments below in Red. We especially appreciate the in-depth questions, that allowed us to reflect on way to improve our system in future applications.

This manuscript has been greatly improved in scope, organization, and readability since the first submission. However, the details of each small experiment sometimes obscure the overall point of the paper, which is to demonstrate that this new vaporization system is a viable unit for field-calibration of atmosphere measurements and laboratory measurements of liquid standards or samples.

We acknowledge the point of the reviewer, but below provide a detailed argument why we suggest to keep the descriptions of the small experiments separate from the general message of the manuscript.

Development of the vaporizer allows for some interesting tests that provide important insights beyond that scope, and those insights are important outcomes of this paper which deserve to be highlighted, but can be distracting from the original point of the paper as they are currently presented. Consider restructuring a bit more to move all of those details to the discussion (e.g., showing that the allan variance performance is adequate for field calibration or lab measurements is a result, but showing how the analysis changes with long data truncation and implications for managing memory when processing data seems like more of a discussion point.) I encourage the authors to carefully reread each section and consider how the tests support use of the vaporization module in lab and field measurements so that they can streamline their message. The science is sound and the technical conclusions drawn about vaporization for CRDS will move these technologies forward, but the paper would be more compelling and easier to read if the message is clearer and the purpose of each of the tests is highlighted more effectively.

We have tried to improve the manuscript by making it more to the reader how the small experiments illustrate the versatility and operation of the instrument. We note in our answers below to the reviewer's comments that we believe that the system presented in the manuscript represents an improvement in the Allan Deviation compared to previously presented systems.

There are still several typos and other sections where clearer language would benefit the reader, many of which I have highlighted line-by-line in the attached document.

Thank you. We have noted this.

Finally, the conversation about system memory requires clarification in several places. More information about how the authors are quantifying the memory of the system and whether or not its performance is adequate for the field-calibration setup or the discrete lab measurements is

needed to show that this system can be used successfully for both functions. The authors in a few places talk about "handling memory" but seem to be referring to post-processing techniques to remove data from very long runs for the purpose of stability analyses, but it is not clear how the memory is treated in day-to-day operations. Can you include a section with general workflow and recommendations for making low-memory measurements and processing the data? The authors also (correctly) point out that the changing flow rates in the system can change the memory effect, but there is no analysis of this impact or best practices for operating a system like this with so many variables. How do you ensure that the memory is adequate for your applications?

We have so far only been using the system in field deployments and for laboratory experiments and not for routine operation of liquid sample measurements.
This means that no stable day-to-day operation has been developed or needed. Instead the way that we are dealing with memory effect is to simply measure the same standard as long time as possible given the constraints of the field deployment or laboratory experiments. For example for laboratory experiments, where we are trying to get 10^-2 order of magnitude d18O precision measurements, we would run the same standard for at least 12 hours.

The authors have not directly compared the data with published values from the continuous-flow systems, but will find that some conclusions drawn from the data truncation exercise are misleading when this is done (figures are overlain in the attachment to demonstrate this). While the allan deviation plots use longer analysis times than previously published data, the discussion of memory and stability improvements isn't very convincing since it seems that the performance is very comparable to published continuous-flow values. The purpose of this analysis seems to be to show that the vaporizer is very stable at long timescales -- and while this is true, it is also demonstrating longer memory length than other systems, which achieve similar values at long timescales without the significant data truncation. The very long allan variance data are important contributions to the literature and the system does show excellent long-term stability – and performance that is on the better end of what other systems have documented – but it seems like the authors want to claim that the performance is "new" or "better" when it both isn't and also doesn't need to be to show that the custom vaporizer unit designed here has excellent performance and unique advantages for its designed application (i.e. humidity-variable calibration). However, the increased memory (especially for dD) should be examined and it needs to be made clear to the reader that this is not an issue for the laboratory measurements or the field calibration that are intended for this unit.

We have address this very important comment in details below.

Please see more specific comments in the attached PDF. Looking forward to reading your updates that will come from this versatile new system!

Technical corrections and minor comments:

ln 12: I suggest rephrasing this to discuss the measurement of many samples instead of their "connection"
We are unfortunately not sure what the reviewer refers to

ln 17: "measuring unknown samples shows" is specific to 17O, right?
Indeed - corrected.

ln 18: the standard error is not provided -- perhaps this is a typo?
In fact - it was our poor sentence structure. It should be clear now.

ln 19: "enhancement" is not quite right here. Maybe rephrase to talk about the increase in deviation or noise level instead?
Corrected

ln 21: typo, should say "factor of 2"
Corrected

ln 27: "achieved to operate" could just say "operated"
Corrected

ln 41: typo after citation, should say "is classically"
Corrected

ln 46-47: please clean up citation formatting and italics. why not use the typical notation for the deuterium excess?
We presume that this will be corrected in the formating stage of the manuscript.

ln 70-73: language is very confusing and inefficient -- please rephrase the first few sentences. A suggestion for ln72 is to say "...deployment is the bubbler system, which has been used continuously..." ln 74: "and that there is minimal" could just say "and the minimal" ln 78: maybe "is not feasible for many campaigns" would be clearer
Corrected

ln 94: double-check the 17O-excess notation here and throughout for consistency
Corrected

ln 97-99: please rewrite this sentence as it is very confusing to follow
Corrected

ln 109: maybe "accuracy, we have further developed the patent application which was published in Steen-Larsen (2016)."
Corrected

ln 115: could say "sufficiently high accuracy for D17O"

Corrected

In 115-130: it might be more intuitive to understand the purpose of the two case studies if this whole section is rewritten as a brief paragraph about objectives and how you tested them
We agree that we could write these bullet points as a paragraph. However, we prefer to keep the bullet points, since we believe that it allow a potential future reader to quickly get an overview of the improvements of the system.

In 135 is a good example of how to refer to the patent for the first time -- consider revising In 109 to be similar
We agree - corrected

In 170: typo, should refer to 4.3. please check section references throughout.
Yes - indeed - corrected and checked.

In 170 and section 4.3 talk about decline in humidity values over long timescales, but figure 5 shows that it sometimes increases. we find that depending on the air flow and water flow, the precipitates either clog the tee itself or the capillary, and depending on which flow is decreasing (air or water) the instability can cause an increase or a decrease in humidity. is this similar? it seems consistent with the values shown in figure 5.
The decline in humidity, which we refer to that are caused by clogging of the capillary is on much longer time-scales that the individual steps shown in figure 5. Typically, we would experience noticible clogging, when we would measure the same humidity level for more than 12 hour periods. We have not had reasons to suspect clogging of the tee itself.

In 194: typo, "between 0.5 to 3.5 should" say "between 0.5 and 3.5"
Corrected

In 195: why not just say "with a 1.59mm PEEK union"?
Corrected

In 220: "frequent" should be "frequently"
Corrected

In 222: SW and WW haven't been defined yet -- please make sure these and other acronyms are defined before use
The acronyms are referring to the standard names in the Table. We will therefore make sure that there is a direct reference to the table. The names of the standards for examples SW and WW was the names, which we were informed about when receiving the standard water from our colleagues.

In 265: please see comments above -- the performance shown here is excellent, but it is not significantly different from Steig et al 2021, which does not trim data as extensively as is shown

here or have nearly as much time for each run. this (and the comparison with fig 8) suggests an increase in memory for this system, which should be discussed in more detail, if only to demonstrate that it can be managed for the applications promised.

Thank you for agreeing that the performance presented is excellent.
While we agree that for integration times up to 10^3 there are only a smaller improvement in our system compared to Steig et al. 2021. The average performance for dD at 10^3 corresponds to our integration at 2e2. However, while the Allan Deviation in Steig et al. 2021 does not improve after 1e3, ours continue to improve until 2e4. See figures below focusing on dD.
The reviewer is however correct in pointing out that the memory effect seems signficantly larger than in the system presented by Steig et al.. Having worked with the similar system system (Jones et al. 2017) as discussed by Steig et al., I do also believe that the system by Jones could have a smaller memory effect than ours. However, it comes at a cost of a 10-20 time higher flow rate.
However, a direct comparison in memory effect between the system used by Steig et al. 2021 and our system is not possible due to lack of information of which standards were measured before the run used to generate the Allan Deviation. We speculate that the Allan Deviation is generated based on the lab water run depicted in Figure 1 and that the run depicted is typical. This means that a idealized step change is equal to about 25 per mil in d18O. This should be compared to the step change of about 55 per mil in d18O applied in our setup.
It is furthermore not clear to us, how much of the data is removed after the step change in Steig et al. in order to remove the effect of memory in the system. In our figure 3 (shown below) we depict the influence of memory effect, when not removing the first 16 hours as dashed lines. Not removing any data after a step change is of course not realistic, but we do this to illustrate the influence of the memory effect on the Allan Deviation.
In short, it would be very interesting to 1) run the system of Jones et al. 2017 for 92 hours continuously and 2) generate a similar plot as Figure 4 based on a step-change of 55 per mil in d18O.

[Figure]

section 3.2 compares the memory for this new system to the memory of the autosampler, which demonstrates that this method could be useful for lab measurements. But how is memory handled for field-calibration? What is a typical field calibration workflow?

Often, for water vapor isotope field measurements, the signal measured is much larger than the measurement uncertainties. This means that less focus is placed on achiving calibrations with precisions achieved only after 1e3 second integrations. However, when measuring water vapor isotopes at low humidity levels, one has to be careful about achiving robust humidity-isotope-calibration of the instrument. To achieve optimal humidity-isotope calibration we would often run the same standard over-night before starting the humidity-isotope-calibration of the instrument.

In 318: should specify that the short-term trend is for the water vapor concentration

Corrected - the text now reads:
As expected, the largest Allan Deviation with averaging time of 600 seconds is the one measured at 584 ppmv level (0.04, 0.05 and 0.17 ‰ for $\delta^{17}O$, $\delta^{18}O$ and $\delta D$, respectively). Between 5000 and 20000 ppmv, the 600 s Allan Deviation is characterized by small variability in general (0.013 ± 0.002, 0.014 ± 0.004 and 0.02 ± 0.01 ‰ for $\delta^{17}O$, $\delta^{18}O$ and $\delta D$, respectively). For unknown reasons the worst performances in terms of Allan Deviation are observed at 11584 ppmv (0.02, 0.03 and 0.06 ‰ for $\delta^{17}O$, $\delta^{18}O$ and $\delta D$, respectively for 600 s averaging time). This is in contrast with the analysis above, which shows the smallest short-term trend of the $H_2O$ signal in the ~12500 ppmv region, which is ~0 ppmv/h.

In 337: it seems like the variability in dD values at low-humidity ranges in Fig. 7 could also be influenced by the relatively longer memory of dD in the system – has this been investigated? Even if the liquid flow rate is the same at all levels because of the secondary mixing at TEE2, the retention time of water within the optical cavity itself should also worsen with decreased humidity which seems like it could contribute to this effect?

We do not believe that the depicted increase in variability of dD at low-humidity levels is a result of memory effect as proposed by the reviewer. We see the increase in variability in dD at low humidity level to be a result of the lower humidity, which gives less signal-noise ratio in cavity. However, we do agree that the hypothesis proposed by the reviewer, that there would be an increase in dD memory effect at low humidity levels due to the fact that there is less molecules to exchange with the molecules adhering to the side of the wall (which could maybe be assumed to a first approximation be independent of the number of molecules in the air).
The hypothesis could be investigated, but it would be useful if such experiment could reveil more information about molecular bonds and adhesion than just a qualitative description.

In 364: many readers will have difficulty understanding the statement of D17O-->d17O conversion
Corrected and simplified.

In 377: consider "values" or something more specific than "one" for clarity
Corrected

In 379: it is unclear what you mean by "error in the scale"

Indeed - changed to "due to measurement error of the weighing scale"

In 384+: this section is describing the experimental setup to examine the high-frequency noise – consider restructuring to include relevant details in the methods section of the paper instead
We agree with the reviewer, that it is possible to move the description of the experimental setup to the method section. In addition, this could also be done for the other application studies. However, below we argue for why we prefer to keep the structure of section as it is now:
1) Our main focus of the paper is to describe how the vapor generation module functions and to document its performance. This section is focused on the origin of the noise generated by the analyzer and is therefore an illustration of what the vapor generation module can be used for. By including the method description of this experimental setup into the method section of the article we feel that the manuscript would loose a bit focus.
2) We want to ensure that it is easy for the reader to build the vapor generation module. It is therefore important for us to keep this focus.

In 392: typo, "exceeding" should be "excess"
corrected

In 410-11: typo? – this sentence doesn't make sense to me
Corrected

In 480-5: this does seem like a general problem for automated adjustments, though certainly not beyond characterization. Have you attempted to characterize the range of this effect for your system?
We agree with the reviewer that this can be characterized. We, however, have not done this yet, as we initially did not plan to log the flow rate of the dilution MFC, which is a requirement for doing so. As we are in the process of building a dedicated line for liquid measurements the characterization of the role of flow rate on memory effect is a natural action item. It is worth to remember that this characterization will be depending on the individual configuration of the system such as tube length.

Ln 484/S3: which standards are used for calibration in each of the tests? It is not clear if it is the same every time and which of the standards listed in the supplement are used.
We have updated the text to make it clear that it is the two standards BER and SP, which is being used.

Ln 491: what is "relatively longer"?
This sentence is focused on explaining the consequences of the different memory effect on dD compared to d18O. The consequence is that one need to have a *relative longer* measurement time for dD compared to the measurement time for d18O to achieve the same percentage of the isotope shift.

Ln 494: is this memory limitation practical? Is it possible to shorten the tubing, or remove other dead volume from the system?

We have not carried out an exhausted search for memory effect removal, but we suspect that the analyzer itself is one of the main drivers of the memory effect. One could in fact try different tube lengths and investigate if the memory effect would be linear and then calculate the inherent memory effect of the system for a zero tube length.

ln 520: relative discrete autosampler injections
We are not sure what the reviewer is referring to here.

ln 536: does not say what the standard error is (and looks copied from earlier section of the paper)
Our text was formulated poorly. This is corrected now.

ln 537-9: citations would be helpful here and elsewhere – this should also be discussed earlier in the paper
We do not have a reference for this speculation as this is something that we have discussed with colleagues. The thinking being that since there are less heavy isotopes in more depleted samples the uncertainty should also be larger. We have changed the text to a hypothesis.

table 1: how is 17Oexcess determined for BER? d17O does not have enough significant digits to determine these values.
The BER standard has been provided with only two significant digits, hence we assumed d17O to be equal to -0.0500 and calculated the 17O Excess.
We are aware that d17O ranging from -0.046 to -0.054 might introduce a 8 per meg variation in 17O excess. However, such a variability in d17O affects only to a lesser extent the slope of the calibration line. Therefore, the effect on the analysis of the Allan variance and liquid measurement reproducibility is not significantly affected.

table 2: I'm not sure there's an advantage to showing both sets of calibrated values -- perhaps just choose one calibration method, explain it in the text, and report the values in the table?

We decided to report the results obtained by the two calibration schemes to demonstrate the good reproducibility of the measurement. Indeed, the independent calibration scheme should be more sensitive to changes in the measurement conditions, while the average calibration scheme should smooth-out the variability more effectively. This effect is visible on the magnitude of the uncertainty associated with the mean of M85, which is larger for the independent calibration. However, the similarity of the results obtained with the two methods provide general evidence that the CRDS drift, the instability of the calibration system and the sample degradation are minimal.